# ENHANCING SPARSE EVENT DETECTION IN HEALTHCARE TIME-SERIES VIA ADAPTIVE GATE OF CONTEXT–DETAIL INTERACTION

**Beomjun Bark**
Seers Technology
pbj7698@gmail.com

**Yun Kwan Kim**[*]
Seers Technology, Korea University
ykwin@korea.ac.kr

## ABSTRACT

Accurate detection of clinically meaningful events in healthcare time-series data is crucial for reliable downstream analysis and decision support. However, most existing methods struggle to jointly localize event boundaries and classify event types; even detection transformer (DETR)-based approaches show limited performance when confronted with extremely sparse events typical of clinical recordings. To address these challenges, we propose a coarse-to-fine detection framework combining a global context explorer, a local detail inspector, and an adaptive gating module (AGM) that fuses multiple label perspectives. The AGM uses transformed labels—encoding event presence and temporal position—to improve learning on sparse events. This design acts as a switch that selectively activates detailed feature extraction only when an event is likely, thereby reducing noise and improving efficiency in sparse settings. We evaluate our framework on diverse healthcare datasets—including arrhythmia detection, emotion recognition, and human-activity monitoring—and demonstrate substantial performance gains over existing DETR-based models, with particularly strong improvements in sparse event detection. With precise and robust event detection, our framework enables interpretation and actionable insights in real-world clinical applications.

## 1 INTRODUCTION

Time-series data are generated across diverse physical, biological, and socio-economic systems, exhibiting characteristics that vary substantially depending on their source. Among these, biosignals—electrical or physical measurements of physiological activity—are particularly distinctive, combining inherent temporal periodicity with irregular, sparse, and clinically significant local events, and they are typically acquired at high sampling rates (Yuan et al., 2023). For instance, arrhythmia episodes in electrocardiograms (ECG) or the onset and offset of specific physiological behaviors must be captured with high temporal precision, as accurate timing is critical for clinical decision-making. Detecting such events therefore requires not only distinguishing normal from abnormal segments but also predicting both the event type (class) and its temporal boundaries—a capability essential for early diagnosis, continuous monitoring, and real-time alert systems (Figure. 1).

Previous studies have explored various approaches for time-series event detection. A common strategy is to leverage unsupervised or semi-supervised anomaly detection methods (Ergen & Kozat, 2019; Braei & Wagner, 2020; Qiao et al., 2024; McIntosh & Albu, 2024), which effectively identify abnormal segments but provide little information about what type of event is occurring. Another line of work frames detection as a classification problem over fixed-length windows (e.g., 10 seconds). These methods predict whether an event occurs within each segment (Bark et al., 2023; Mandala et al., 2025; Saranya et al., 2025), particularly in studies involving biosignal-based events or disease detection. However, such window-based formulations cannot capture exact onset and offset times, limiting their ability to perform boundary-aware event detection within fixed segments.

---

[*]Corresponding author.

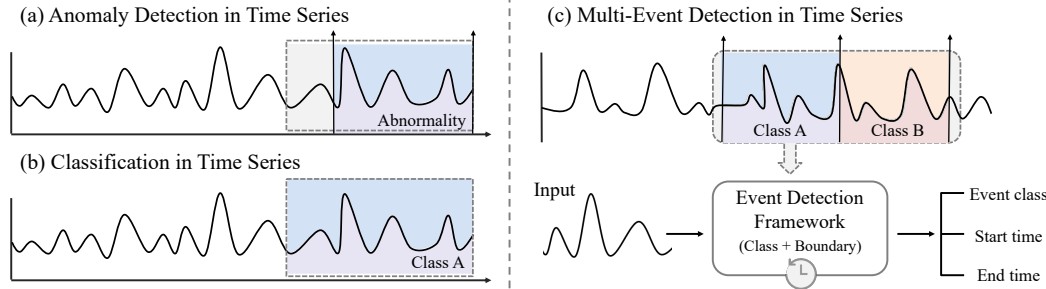

Figure 1: (a) Anomaly detection identifies abnormal segments without event types. (b) Classification labels fixed-length time windows, without explicit boundary localization. (c) Multi-event detection predicts both event class and temporal boundaries within each segment, enabling accurate detection.

Overall, existing approaches face two key challenges: (1) learning the type of events under sparse conditions, and (2) accurately localizing event boundaries. In clinical applications, these challenges are particularly critical, as failure to identify the onset, offset, and type of events can prevent timely interventions. Moreover, providing explicit temporal and class information allows clinicians to focus on specific suspicious segments rather than reviewing entire recordings, thus improving workflow efficiency, supporting evidence-based decision-making, and generating actionable insights. Consequently, previous methods are limited in their ability to perform event detection in time-series data, highlighting the need for accurate, boundary-aware detection that facilitates both reliable interpretation and clinical actionability.

To address these challenges, we propose a novel framework for time-series data composed of two key units: (i) an adaptive gating module that facilitates dynamic interaction between global and local features, and (ii) a coarse-to-fine global–local structure built upon the detection transformer (DETR) architecture (Carion et al., 2020; Sarlin et al., 2019). This design jointly predicts event types and their temporal boundaries, enabling effective learning from sparse events. During training, the model leverages multiple label perspectives—including actual values, event existence, and occurrence locations—which allows robust detection even under extreme event sparsity. We validate our approach on healthcare time-series datasets, including arrhythmia detection, emotion recognition, and activity monitoring, demonstrating consistent performance across diverse multi-event detection tasks.

The main contributions of this work are summarized as follows:

- We propose a time-series multi-event detection framework based on DETR that simultaneously estimates event types and temporal boundaries, enabling effective interaction between contextual and detailed features.

- We introduce an adaptive gating module that enables effective detection of sparse events by transforming labels to emphasize different aspects—such as event existence or occurrence locations—allowing the model to learn complementary temporal cues.

- We demonstrate the clinical applicability of our method through diverse experiments on healthcare datasets, showing its effectiveness for real-world multi-event detection tasks.

## 2 RELATED WORK

### 2.1 SEQUENCE-TO-SEQUENCE MODELS FOR EVENT DETECTION

Seq-to-seq models for time-series analysis have evolved rapidly, starting from LSTM (Graves, 2012), TCN (Bai et al., 2018), and Inception Time (Ismail Fawaz et al., 2020), and more recently advancing to Transformer-based architectures (Vaswani et al., 2017; Nie et al., 2022; Liu et al., 2023) and time-series–specific foundation models (Garza et al., 2023; Das et al., 2024). While these models are effective at capturing global patterns and performing point-wise prediction or regression, they face structural limitations for event boundary detection within segments. Point-wise outputs are inherently insufficient for tasks that require identifying both event onset and offset. Moreover, these models are not well-suited for detecting sparse events, where occurrences are rare and irregular.

## 2.2 END-TO-END DETR MODELS FOR EVENT DETECTION

Several approaches have been proposed for event detection, among which DETR (Carion et al., 2020) has become one of the most widely adopted and influential architectures. DETR introduced the first end-to-end transformer-based framework for object detection, replacing traditional anchor-based pipelines. Subsequent variants, such as Deformable DETR (Zhu et al., 2020), DINO[1] (Zhang et al., 2022), and DEIM[2] (Huang et al., 2025), improved both convergence speed, accuracy, and training efficiency. However, these advances have largely been limited to the image domain. In time-series data, most timestamps correspond to normal states, with events occurring sparsely and exhibiting ambiguous temporal boundaries (Azib et al., 2023; Zamanzadeh Darban et al., 2024). These characteristics introduce a significant mismatch between conventional DETR architectures and the requirements of boundary-aware event detection in time-series data. Direct application of DETR often yields inaccurate predictions and inefficient training under extreme sparsity.

## 3 METHODOLOGY

Our framework processes univariate or multivariate time-series data using an end-to-end encoder–decoder architecture. The components of the proposed framework and their operating principles, including the overall weight flow, are described below. A schematic overview of the full framework is shown in Figure 2. The input and output are defined as:

$$X \in \mathbb{R}^{B \times T \times D} \quad \longmapsto \quad Y = \{(c_i, t_i^{\mathrm{on}}, t_i^{\mathrm{off}})\}_{i=1}^N, \tag{1}$$

where $B$ is the batch size, $T$ is the segment length, and $D$ is the number of variables ($D = 1$ for univariate, $D > 1$ for multivariate time-series data). The output $Y$ is a set of $N$ events, where $c_i$ denotes the event class, and $t_i^{\mathrm{on}}, t_i^{\mathrm{off}}$ represent the onset and offset times of the $i$-th event, respectively.

### 3.1 ARCHITECTURE OVERVIEW

We use a frozen foundation model (FM), denoted $f_{\mathrm{FM}}$, based on the Chronos-T5 tiny-bolt architecture (Ansari et al., 2024) pre-trained on large-scale time-series data. The FM serves as a lightweight backbone. Since Chronos-T5 was originally designed for univariate time series, the framework employs a flexible input structure that expands or contracts the FM according to the input dimensionality, allowing it to process multivariate time-series data. For a multivariate input, the representation is constructed as

$$z = \bigoplus_{j=1}^D f_{\mathrm{FM}}^{(j)}(X_{:,j}) \in \mathbb{R}^{B \times \tau \times (D \cdot d_{\mathrm{FM}})}, \tag{2}$$

where $\bigoplus$ denotes channel-wise concatenation, $B$ is the batch size, $X_{:,j}$ is the $j$-th input channel, $\tau$ is the number of time steps in the FM output, and $d_{\mathrm{FM}}$ is the embedding dimension of a single channel.

The embedding vector $z$ extracted from the FM($f_{\mathrm{FM}}$) is first mapped to an integrated representation $h \in \mathbb{R}^{B \times \tau \times d}$ through a feed-forward network (FFN) comprising linear layers and positional embeddings. The representation $h$ then provides the foundational input to two feature extraction modules: the global context explorer (GCE) and the local detail inspector (LDI), which extract global and local temporal features, respectively, each with its own positional embeddings. Since GCE and LDI follow different processing paths, their outputs may reside in distinct feature spaces; an alignment layer is used to map all features into a common space.

Next, the core component of the framework, the adaptive gating module (AGM), fuses the aligned features by modulating their contributions using a conditional gate scaler (CGS) and positional Gaussian injection (PGI), based on the global and local distributions of each sample. Specifically, the AGM computes gate values for each time step, controlling the relative contributions of GCE and LDI features. Finally, the gated features are passed to the transformer decoder along with a set of learnable object queries $Q = \{q_1, q_2, \ldots, q_N\}$ to predict both the start and end times of events as well as their class labels for each segment.

---

[1]DINO: DETR with Improved DeNoising Anchor Boxes for End-to-End Object Detection
[2]DEIM: DETR with Improved Matching for Fast Convergence

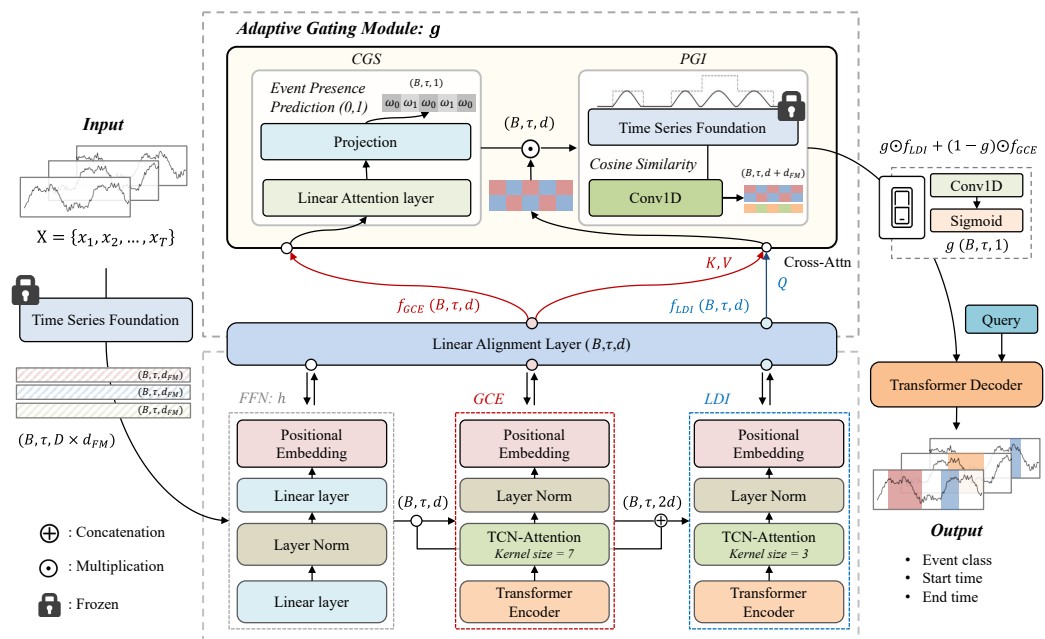

Figure 2: Overview of the proposed framework. The input time-series data is first processed by a frozen foundation model (FM), followed by a feed-forward network (FFN). Global and local temporal features are then extracted via the global context explorer (GCE) and local detail inspector (LDI), respectively. The GCE and LDI outputs are fused through the adaptive gating module (AGM), which acts as a dynamic gate to modulate global and local information. The fused representation is subsequently fed into the transformer decoder to predict event types and their temporal boundaries.

## 3.2 GLOBAL CONTEXT EXPLORER & LOCAL DETAIL INSPECTOR

The model operates on three complementary inputs: (i) the initial feature representation $h$, (ii) the global context extracted by the GCE, and (iii) the local detail extracted by the LDI. The GCE ($f_{\text{GCE}} \in \mathbb{R}^{B \times \tau \times d}$) and LDI ($f_{\text{LDI}} \in \mathbb{R}^{B \times \tau \times d}$) are composed of a similar block structure comprising a transformer encoder, a TCN-based attention mechanism with dilation rates $[1, 2, 4, 8]$, and layer normalization, providing complementary perspectives for event representation in time series. To ensure consistency across these heterogeneous inputs, all three inputs are projected into a unified representation space through a linear alignment layer. Specifically, the initial features are aligned as

$$h_{\text{align}} = \text{Align}(h), \tag{3}$$

while the outputs of GCE and LDI are also maintained in the aligned space, which normalizes scale and distribution across inputs, unifies feature dimensions for consistent integration, and facilitates semantic alignment between global and local features, inspired by (Lin et al., 2017).

Building on this aligned representation, the coarse-to-fine hierarchy is established not only through the different kernel sizes (7 for GCE and 3 for LDI) but also via two additional factors. First, LDI receives both the aligned input and the globally refined GCE output:

$$x_{\text{LDI}} = h_{\text{align}} \oplus f_{\text{GCE}}(h_{\text{align}}), \tag{4}$$

creating a richer, higher-dimensional space that enhances its ability to capture fine-grained local interactions. Second, the sequential ordering of the modules—GCE preceding LDI—enables GCE to extract coarse, long-range dependencies, after which LDI focuses on refining these representations with local detail.

Because all features remain in the aligned space, downstream components such as the AGM can consistently integrate global and local information. This hierarchical, aligned processing is critical for accurate, boundary-aware event detection and stable model training.

Figure 3: Overview of the adaptive gating module (AGM). (a) Conditional gate scaler (CGS) mitigates class imbalance by predicting weights $w_c$ to emphasize sparse event features while downscaling frequent ones. (b) Positional Gaussian injection (PGI) encodes event locations by aligning Gaussian-distributed labels with input features via cosine similarity.

## 3.3 ADAPTIVE GATING MODULE

The AGM is a core component of our framework (Figure. 3), consisting of two sub-modules: conditional gate scaler (CGS), which adaptively modulates the contributions of each feature stream to better capture events in time series, and positional Gaussian injection (PGI), which injects positional priors to enhance temporal localization of events. The input to the AGM is the fused representation from GCE and LDI, formulated as

$$x_{\text{AGM}} = \text{CrossAttn}(f_{\text{GCE}}, f_{\text{LDI}}), \quad x_{\text{AGM}} \in \mathbb{R}^{B \times \tau \times d}, \tag{5}$$

Within the AGM, $x_{\text{AGM}}$ sequentially passes through the CGS and PGI submodules. The resulting output, denoted as $\text{output}_{\text{PGI}}$, is then compressed through a 1D convolution layer followed by a sigmoid activation to generate the gate tensor:

$$g = Sigmoid(\text{Conv1D}(\text{output}_{\text{PGI}})), \quad g \in \mathbb{R}^{B \times \tau \times 1}. \tag{6}$$

This gate tensor dynamically modulates the relative contributions of LDI and GCE via element-wise weighting:

$$h_{\text{gated}} = g \odot f_{\text{LDI}} + (1 - g) \odot f_{\text{GCE}}, \quad \odot: \text{ element-wise multiplication.} \tag{7}$$

In summary, CGS and PGI work in sequence to produce $g$, enabling the AGM to adaptively balance global and local information while preserving boundary precision for sparse event detection.

### 3.3.1 CONDITIONAL GATE SCALER

The CGS is designed to enhance the learning of sparse events by modulating the relative importance of features. It reflects distributional differences between event and non-event segments, assigning higher weights to event-related representations. In particular, CGS controls how much the LDI can leverage the global context extracted by the GCE, and therefore only $f_{\text{GCE}}(h_{align})$ is used as input.

For each input segment, we reinterpret the labels from a binary perspective (*no event*: 0, *event*: 1) and simultaneously assign a weight in $(0, 1)$ to mitigate imbalance. Specifically, let $r_0$ and $r_1$ denote the ratios of the two classes (e.g., 10:1). The GCE features are fed through an FFN to predict the binary label, supervised by a weighted cross-entropy loss $\mathcal{L}_{\text{BCE}}$, with class weights computed as:

$$w_0 = \frac{1/r_0}{1/r_0 + 1/r_1}, \quad w_1 = \frac{1/r_1}{1/r_0 + 1/r_1}, \tag{8}$$

ensuring $0 < w_0, w_1 < 1$, with the rarer class receiving the larger weight. The final output of the FFN serves as the conditional scaling weight $w_c$, directly modulating the AGM input.

Specifically, for each sample, the AGM input $x_{\text{AGM}}$ is element-wise scaled by this conditional weight as $\tilde{x}_{\text{AGM}} = w_c \odot x_{\text{AGM}}$, where $\tilde{x}_{\text{AGM}} \in \mathbb{R}^{B \times T \times d}$. As a result, abundant (non-event) features are downscaled while sparse (event) features are emphasized, enabling adaptive rebalancing during training.

### 3.3.2 POSITIONAL GAUSSIAN INJECTION

The PGI module encodes event locations within a time series by introducing Gaussian-shaped supervisory signals. For each ground-truth event, we generate a normalized Gaussian label distribution ($y_{\text{gaussian}}$) centered at the event midpoint, with the start and end points explicitly set to zero to delineate boundaries:

$$
y_{\text{gaussian}}(t) = \begin{cases} \dfrac{\mathcal{N}(t; c, \sigma^2)}{\max_{u \in [t_s, t_e]} \mathcal{N}(u; c, \sigma^2)}, & t_s < t < t_e, \\ 0, & t \in \{t_s, t_e\}, \end{cases} \tag{9}
$$

where $t_s$ and $t_e$ denote the start and end indices of an event defined by $y(t) \neq 0$, $c = \frac{t_s + t_e - 1}{2}$ is the event center, and $\sigma$ is proportional to the event length. Here, $\mathcal{N}(t; c, \sigma^2)$ represents the value of a 1D Gaussian distribution with mean $c$ and variance $\sigma^2$. The normalization by the maximum value ensures the label distribution peaks at 1 within the event.

This Gaussian supervision is deliberately designed to: (i) explicitly indicate sparse event centers, (ii) enhance the distinguishability of consecutive events, and (iii) prevent boundary ambiguity that arises when events are split across fixed-length segments. In this way, PGI provides the model with smooth boundary cues related to event onsets and offsets within each segment. It thus complements the original discrete label representation, offering a richer supervisory signal for event localization.

To incorporate this signal, input features to the AGM after CGS ($\tilde{x}_{\text{AGM}} \in \mathbb{R}^{B \times \tau \times d}$) are passed through a trainable convolutional layer ($d \to d_{\text{FM}}$), while Gaussian labels are projected via the frozen FM ($f_{\text{FM}} \in \mathbb{R}^{B \times \tau \times d_{\text{fm}}}$) encoder into the same representation space. A cosine similarity loss aligns the two:

$$
\mathcal{L}_{\text{cos}} = 1 - \frac{1}{B\tau} \sum_{b=1}^{B} \sum_{t=1}^{\tau} \frac{\text{Conv}(\tilde{x}_{\text{AGM}})_{b,t} \cdot f_{\text{FM}}(y_{\text{gaussian}})_{b,t}}{\|\text{Conv}(\tilde{x}_{\text{AGM}})_{b,t}\|_2 \, \|f_{\text{FM}}(y_{\text{gaussian}})_{b,t}\|_2}, \tag{10}
$$

where $B$ is the batch size and $\tau$ the time-step length. The final PGI output is obtained by concatenating the AGM input after CGS with its convolutional layer along the feature dimension,

$$
\text{Output}_{\text{PGI}} = \tilde{x}_{\text{AGM}} \oplus \text{Conv}(\tilde{x}_{\text{AGM}}) \in \mathbb{R}^{B \times \tau \times (d + d_{\text{FM}})} \tag{11}
$$

### 3.4 TRAINING OBEJECTIVE

Our framework is trained with three loss terms: (1) a cosine similarity loss ($L_{\text{cos}}$) between the Gaussian labels generated by the PGI module and the AGM convolutional outputs, (2) a binary cross-entropy loss ($L_{\text{BCE}}$) applied to the CGS predictions, and (3) a detection loss ($L_{\text{Detection}}$) in the transformer decoder, computed via Hungarian matching. The overall objective is given by

$$
L_{\text{total}} = \alpha L_{\text{cos}} + \beta L_{\text{BCE}} + \gamma L_{\text{Detection}}, \quad \alpha = 0.2, \ \beta = 0.1, \ \gamma = 0.7. \tag{12}
$$

Unlike the DETR, where Hungarian matching is based on 2D bounding box position and size, temporal event detection requires a tailored cost. We therefore define the matching cost as

$$
\text{cost}(i, j) = \lambda_{\text{cls}} \cdot \text{cost}_{\text{cls}}(i, j) + \lambda_{\text{ctr}} \cdot L_1(c_i, c_j) + \lambda_{\text{len}} \cdot L_1(l_i, l_j), \tag{13}
$$

where $c_i, c_j$ denote event centers and $l_i, l_j$ their lengths. We set $\lambda_{\text{cls}} : \lambda_{\text{ctr}} : \lambda_{\text{len}} = 1 : 5 : 1$.

The detection loss ($L_{\text{Detection}}$) combines localization and classification terms, formulated as $L_{\text{Detection}} = 5.0 \cdot L_{\text{loc}} + 2.0 \cdot L_{\text{cls}}$, where $L_{\text{loc}}$ is an $L_1$ loss over centers and lengths, and $L_{\text{cls}}$ is a weighted cross-entropy loss. To mitigate class imbalance, the classification weights are assigned inversely proportional to class frequencies, emphasizing sparse event classes while down-weighting abundant normal segments.

# 4 EXPERIMENT

## 4.1 PERFORMANCE COMPARISON ON HEALTHCARE DATASETS

**Datasets** We evaluate our model on four public databases covering arrhythmia detection, emotion recognition, and activity monitoring. Specifically: MIT-BIH (Moody & Mark, 2001) and SHDB-AF (Tsutsui et al., 2025) for arrhythmia detection, WESAD (Schmidt et al., 2018) for emotion recognition, and OPP (Roggen et al., 2010) for activity monitoring. For arrhythmia detection, we consider both specific arrhythmias (MIT-BIH Class 3, SHDB-AF Class 3, MIT-BIH Class 15)and broader categories (MIT-BIH Class 6, SHDB-AF Class 5). All datasets are split into training, validation, and test sets (8:1:1). Sampling frequency and channel count vary: MIT-BIH and SHDB-AF are 256 Hz with 1 and 2 channels; WESAD is 200 Hz with 8 channels; OPP is 30 Hz with 36 channels. Time-series signals are first resampled, then interpolated to fill missing values, and finally segmented into fixed-length windows. Stratified sampling is applied to address class imbalance. Pre-processing details and class definitions are provided in *Appendix A*.

**Baseline** We compare our approach against six DETR-based baselines. We employ the same Chronos-T5 model as a pre-trained, fixed feature extractor (backbone encoder) to adapt them to 1D time-series inputs. The extracted embedding vectors are fed into the Transformer encoder of the respective DETR-style architectures, using a 1D structure. Temporal coordinates (event center and length) are used to align with the time-series data. All models, including DETR (Carion et al., 2020), Multi-scale DETR, Deformable-DETR (Zhu et al., 2020), DAB-DETR (Liu et al., 2022a), DN-DETR (Li et al., 2022), and Deformable-DINO (Zhang et al., 2022), were trained for 100 epochs with a batch size of 64, early stopping after five epochs without improvement, and a learning rate of $1 \times 10^{-4}$ using the AdamW optimizer (weight decay $5 \times 10^{-2}$, $\epsilon = 1 \times 10^{-8}$, $\beta = (0.9, 0.999)$). Implementation details are provided in *Appendix C*.

**Metrics** We evaluate performance using three complementary metrics: macro point-wise F1 (PW-F1), macro affiliation F1 (AF-F1), and mean average precision (mAP). PW-F1 measures frame-level accuracy by converting predictions and ground-truth events into time-indexed label sequences. AF-F1 assesses event-level correctness by checking whether each predicted segment overlaps with a ground-truth segment of the same class. Finally, mAP summarizes precision–recall behavior based on confidence scores, treating each event as a temporal segment analogous to DETR-style detection. Together, these metrics capture timing accuracy (PW-F1), segment alignment (AF-F1), and confidence-based detection quality (mAP). Detailed metric definitions are in *Appendix B*.

Table 1: Overall performance comparison across multiple datasets. **Bold** indicates the best performance, while underlined values indicate the second-best performance. Full quantitative results are in *Appendix F*, and qualitative examples of detected events are shown in *Appendix E*.

| Model | Metric | MIT-BIH | | | SHDB-AF | | WESAD | OPP |
|---|---|---|---|---|---|---|---|---|
| | | Class 3 | Class 6 | Class 15 | Class 3 | Class 5 | Class 8 | Class 5 |
| DETR | PW-F1 | 77.22 | 64.41 | 41.32 | 93.24 | 60.19 | 59.53 | 61.30 |
| | AF-F1 | 82.72 | 58.29 | 43.35 | 93.47 | 60.83 | 62.29 | 61.89 |
| | mAP | 51.45 | 53.33 | 45.23 | 95.78 | 77.77 | 53.14 | 50.85 |
| Multi-scale DETR | PW-F1 | 75.60 | 73.36 | 65.52 | 91.99 | 63.49 | 63.88 | 58.05 |
| | AF-F1 | 76.00 | **62.39** | 49.34 | 92.13 | 64.67 | 65.65 | 58.38 |
| | mAP | 66.03 | 55.60 | 50.97 | 97.76 | 76.85 | 68.24 | 48.28 |
| Deformable DETR | PW-F1 | 85.39 | 68.68 | 58.96 | 90.86 | 53.33 | 62.69 | 61.05 |
| | AF-F1 | 84.71 | 60.81 | 45.67 | 91.10 | 53.99 | 65.45 | 60.92 |
| | mAP | 63.17 | 52.59 | **52.16** | 97.79 | 74.18 | 66.59 | 51.86 |
| DAB-DETR | PW-F1 | 83.13 | 71.05 | 53.50 | 93.11 | 51.52 | 55.59 | 60.35 |
| | AF-F1 | 83.70 | 60.77 | 46.03 | 93.23 | 52.14 | 58.58 | 60.88 |
| | mAP | 65.57 | 54.19 | 46.85 | **98.96** | **86.16** | **72.75** | 50.30 |
| DN-DETR | PW-F1 | 77.82 | 66.82 | 62.17 | 91.59 | 71.92 | 66.44 | 57.00 |
| | AF-F1 | 77.71 | 58.18 | 47.37 | 86.26 | 70.13 | 66.32 | 56.05 |
| | mAP | 66.28 | 51.42 | 48.80 | 96.69 | 80.06 | 55.12 | 49.29 |
| Deformable-DINO | PW-F1 | 86.41 | 74.11 | 63.49 | 89.57 | 66.03 | 69.22 | 56.80 |
| | AF-F1 | 83.22 | 60.00 | 48.59 | 88.82 | 65.98 | 70.73 | 55.80 |
| | mAP | 72.05 | 52.20 | 46.92 | 96.00 | 83.65 | 69.96 | 50.68 |
| Ours | PW-F1 | **90.63** | **83.37** | **74.86** | **96.23** | **83.41** | **73.59** | **64.98** |
| | AF-F1 | **85.96** | 60.87 | **52.85** | **96.09** | **83.78** | **74.29** | **62.81** |
| | mAP | **77.66** | **57.54** | 44.55 | 97.38 | 85.83 | 65.19 | **60.07** |

**Results** Table 1 shows that our model consistently outperforms all baselines on both PW-F1 and AF-F1 while maintaining competitive mAP. On MIT-BIH Class 3, our model achieves a PW-F1 of 90.63, exceeding other methods by +4.22–15.03 percentage points (%p). On SHDB-AF Class 5, it reaches an AF-F1 of 83.78, surpassing baselines by +13.65–31.64 %p. Although PW-F1 and AF-F1 are often conflicting, our model attains strong performance on both metrics while sustaining high mAP, indicating that it is not tuned to a single event type but captures both the temporal precision and continuity of diverse events.

Table 2: Summary of notable results for specific dataset events. Each row reports the performance for a specific event within each dataset and provides the corresponding ratio. **Bold** indicates the best performance, while underlined values indicate the second-best performance. Full results for all classes are provided in *Appendix F*.

| Dataset (Event) | Ratio (%) | Metric | DETR | Multi-scale DETR | Deformable DETR | DAB-DETR | DN-DETR | DINO ‖ | Ours |
|---|---|---|---|---|---|---|---|---|---|
| MIT-BIH Class 3 (AFL) | 0.91 | PW-F1 | 64.02 | 58.11 | 77.17 | 72.30 | 66.65 | 78.01 | **84.58** |
| | | AF-F1 | 77.39 | 60.51 | 81.90 | 77.39 | 71.04 | 79.13 | **84.30** |
| MIT-BIH Class 15 (T) | 1.32 | PW-F1 | 0.60 | 38.31 | 34.19 | 28.06 | 58.97 | 60.94 | **80.17** |
| | | AF-F1 | 33.33 | 44.44 | 30.00 | 32.26 | 52.94 | 58.33 | **78.57** |
| SHDB-AF Class 5 (PAT&NOD) | 0.03 | PW-F1 | 29.12 | 37.80 | 22.51 | 12.50 | 43.84 | 25.85 | **66.41** |
| | | AF-F1 | 29.75 | 38.65 | 22.88 | 12.50 | 37.69 | 24.54 | **67.26** |
| WESAD Class 8 (Task 1) | 0.95 | PW-F1 | 30.11 | 31.36 | 10.41 | 37.46 | 41.56 | 49.59 | **60.00** |
| | | AF-F1 | 34.99 | 31.69 | 12.57 | 44.40 | 41.51 | 50.97 | **57.12** |
| OPP Class 5 (Stand) | 42.36 | PW-F1 | 46.75 | 40.02 | 36.45 | 44.28 | 35.35 | 35.73 | **55.61** |
| | | AF-F1 | 45.91 | 40.61 | 32.97 | 44.53 | 27.37 | 29.18 | **48.83** |
| MIT-BIH Class 6 (Ventricular arrhythmia) | 0.59 | PW-F1 | 20.26 | 44.59 | 29.88 | 44.51 | 24.53 | 54.71 | **79.04** |
| | | AF-F1 | 28.27 | **36.36** | 30.83 | 25.27 | 21.75 | 25.60 | 19.52 |

**Highlights 1** Table 2 highlights the distinctive strengths of our model on selected classes. Even for extremely sparse events, our method outperforms strong baselines, achieving combined PW-F1 and AF-F1 improvements over Deformable-DINO ranging from 6.6 to 20.2 %p. Performance on non-sparse events (e.g., OPP Class 5 "Stand") is also competitive. For challenging cases such as "Ventricular arrhythmia" in MIT-BIH Class 6, our model attains high PW-F1 but lower AF-F1. This indicates that while the precise boundaries of individual events may not always be perfectly captured, the model reliably detects event occurrence—a property particularly valuable in healthcare applications.

Table 3: Summary of model performance by event length. Event classes are positioned as short or long based on their class-specific mean event length (CMEL, in seconds) relative to the global mean event length (GMEL, in seconds), considering a fixed window length. This table provides a detailed comparison of performance metrics for cases with short or long events across multiple datasets, based on the GMEL. **Bold** indicates the best performance, while underlined values indicate second-best performance. Full results for all classes are provided in *Appendix F*.

| Dataset (Event) | CMEL (GMEL) | Metric | DETR | Multi-scale DETR | Deformable DETR | DAB-DETR | DN-DETR | DINO ‖ | Ours |
|---|---|---|---|---|---|---|---|---|---|
| MIT-BIH Class 3 (AFIB) | 8.52 (8.32) | PW-F1 | 90.42 | 93.08 | 93.61 | 93.95 | 88.99 | 94.81 | **96.67** |
| | | AF-F1 | 88.04 | **91.49** | 87.51 | 90.00 | 84.44 | 87.31 | 87.91 |
| MIT-BIH Class 15 (SVTA) | 7.75 (5.19) | PW-F1 | 81.21 | **96.71** | 94.72 | 96.36 | 94.24 | 96.43 | 96.34 |
| | | AF-F1 | 84.53 | 85.71 | 85.42 | 88.42 | 85.22 | 87.11 | **88.45** |
| MIT-BIH Class 15 (P) | 2.95 (5.19) | PW-F1 | 6.63 | 69.51 | 71.55 | 63.31 | 70.26 | 70.60 | **80.05** |
| | | AF-F1 | 37.60 | 44.64 | 50.83 | 51.91 | 50.25 | 51.15 | 55.79 |
| MIT-BIH Class 15 (VFL) | 0.88 (5.19) | PW-F1 | 3.36 | 31.54 | 31.53 | 22.68 | 16.55 | 32.16 | **69.03** |
| | | AF-F1 | 15.37 | 17.43 | 13.27 | **22.93** | 9.38 | 11.69 | 10.27 |
| SHDB-AF Class 5 (AT) | 9.61 (9.90) | PW-F1 | 42.69 | 58.20 | 38.28 | 32.81 | 65.82 | 55.68 | **75.20** |
| | | AF-F1 | 43.59 | 59.29 | 38.81 | 33.85 | 66.23 | 57.64 | **76.16** |
| WESAD Class 8 (Stress) | 9.45 (9.41) | PW-F1 | 75.93 | 84.41 | 85.44 | 83.63 | 77.02 | 85.28 | **86.58** |
| | | AF-F1 | 77.45 | 84.94 | 85.79 | 83.70 | 73.76 | 85.33 | **86.57** |
| OPP Class 5 (Lie) | 6.80 (3.93) | PW-F1 | 83.46 | 76.22 | 84.78 | 79.69 | 81.31 | 82.58 | **86.67** |
| | | AF-F1 | 85.58 | 79.13 | 87.03 | 82.54 | 85.03 | 85.27 | **87.86** |

**Highlights 2** Table 3 emphasizes the model's performance with respect to relative event length. To account for class-dependent duration characteristics and to provide an objective, sparsity-independent evaluation, we adopt CMEL and GMEL as length-aware metrics. The results show that the model generally achieves high performance regardless of event length. This trend is particularly evident in the MIT-BIH Class 15 dataset, where the model outperforms baselines for both the SVTA class—whose event duration exceeds the GMEL—and the P class, which is shorter than the

GMEL. A notable exception is the VFL class, which contains extremely short events. Although the model exhibits outstanding PW-F1 performance (exceeding the previous best by 36%p), its AF-F1 performance is lower than that of the comparison models. This suggests that while our model can accurately identify fine-grained points within very short events, its ability to detect an entire event instance as a unified and independent segment still leaves room for improvement.

## 4.2 ANALYSIS OF INTERACTION BETWEEN GCE AND LDI

**Motivation** To investigate how our model dynamically integrates multi-scale information, we analyzed interactions between the GCE and LDI under event and non-event conditions. High-dimensional outputs were aggregated via mean and normalized for visualization (Figure. 4).

**Observation** Figure 4 illustrates the dynamic behavior of the AGM. In stable conditions without events, the AGM output closely follows the coarse-grained GCE values, indicating reduced reliance on computationally expensive fine-grained representations such as LDI and primarily leveraging stable global information for efficiency. In contrast, when events occur, fluctuations in both LDI and GCE become pronounced, and the model switches to a GCE-LDI interaction strategy to accurately capture subtle changes and utilize rich LDI representations for improved predictive performance.

**Insights** These results highlight how the AGM leverages context-detail interactions to adaptively select feature scales: emphasizing efficiency by relying on coarse-grained context in stable periods, while exploiting fine-grained detail to maximize sensitivity during critical events. This adaptive strategy is particularly effective for sparse event detection in healthcare time-series data.

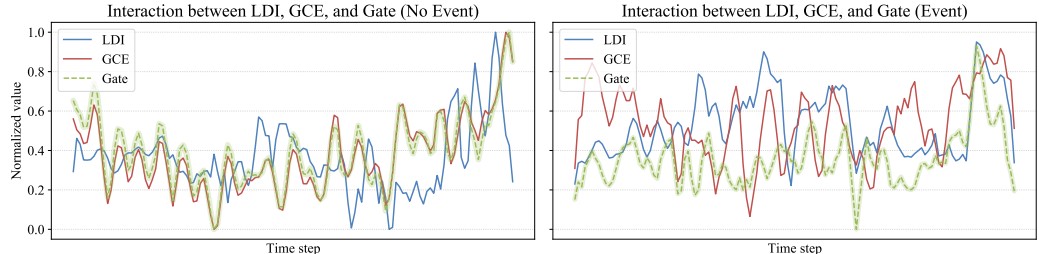

Figure 4: Adaptive gating module selectively prioritizes feature scales. The normalized outputs of LDI, GCE, and AGM are shown over time. In non-event periods (left), the gate closely follows the low-variability GCE, emphasizing stable global information. During events (right), the gate increases the contribution of high-variability LDI, capturing subtle changes and improving robustness.

## 4.3 ABLATION STUDY

**Quantitative Analysis** We evaluate the contribution of the AGM and its two submodules—the PGI and the CGS—through a joint quantitative and convergence analysis. Under three ablation settings (removing AGM entirely, removing CGS, or removing PGI), the full model consistently achieves the best performance across datasets and metrics (Table 4). Removing the AGM entirely leads to substantial performance drops in both PW-F1 and AF-F1, highlighting the critical role of adaptive gating in integrating multi-scale temporal cues. The PGI-only and CGS-only variants maintain moderate performance, yet they exhibit clear degradation compared to the full model, indicating that each submodule contributes unique, complementary information. Overall, these results quantitatively confirm that the AGM effectively integrates complementary label perspectives and significantly enhances event detection performance.

**Interpretive Analysis** To further understand their roles during training, we analyzed epoch-wise trends (Figure. 5). The CGS-only model converges rapidly in early epochs by emphasizing event presence, whereas the PGI-only model, although slower to start, steadily refines predictions through positional information and eventually surpasses the CGS-only variant after about 15 epochs, achieving higher final performance. This complementary behavior—CGS for early learning and PGI for fine-grained refinement, with PGI ultimately yielding stronger results—explains the superior performance of the full model. Convergence curves of DETR-based baselines are provided in *Appendix D*.

Table 4: Ablation study of the AGM and its components. "w/o" denotes "without." **Bold** indicates the best performance, while underlined values indicate the second-best performance.

| Model | Metric | MIT-BIH | | | SHDB-AF | | WESAD | OPP |
|---|---|---|---|---|---|---|---|---|
| | | Class 3 | Class 6 | Class 15 | Class 3 | Class 5 | Class 8 | Class 5 |
| w/o AGM | PW-F1 | 49.97 | 76.46 | 67.63 | 91.98 | 60.79 | 60.51 | 51.36 |
| | AF-F1 | 55.21 | 51.25 | 39.32 | 91.90 | 61.20 | 63.39 | 46.27 |
| | mAP | 64.31 | 52.69 | 43.44 | 96.55 | 72.69 | 62.30 | 50.12 |
| w/o CGS, with PGJ | PW-F1 | 84.41 | 81.22 | 72.76 | 95.77 | 70.88 | 61.84 | 52.46 |
| | AF-F1 | **87.07** | 59.98 | 45.65 | 95.52 | 79.78 | 63.43 | 45.12 |
| | mAP | 71.46 | 55.41 | 45.50 | 96.57 | 73.06 | 65.15 | 52.25 |
| w/o PGJ, with CGS | PW-F1 | 84.68 | 80.02 | 74.72 | 95.13 | 73.17 | 61.67 | 54.78 |
| | AF-F1 | 82.93 | 58.23 | 44.81 | 95.02 | 73.41 | 63.71 | 46.67 |
| | mAP | 71.69 | 54.89 | **45.68** | 96.53 | 70.68 | 63.55 | 53.31 |
| Ours (Baseline) | PW-F1 | **90.63** | **83.37** | **74.86** | **96.23** | **83.41** | **73.59** | **64.98** |
| | AF-F1 | 85.96 | **60.87** | **52.85** | **96.09** | **83.78** | **74.29** | **62.81** |
| | mAP | **77.66** | **57.54** | 44.55 | **97.38** | **85.83** | **65.19** | **60.07** |

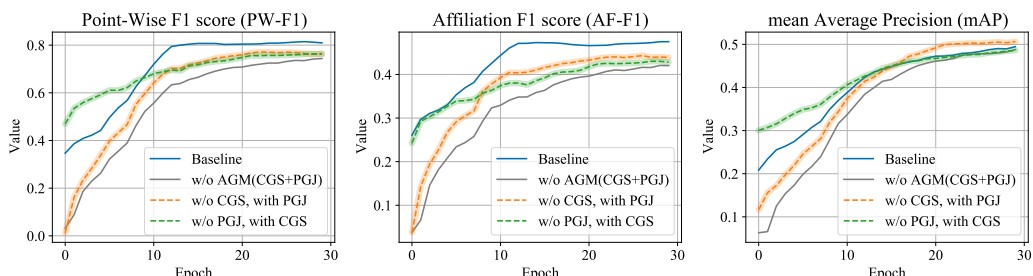

Figure 5: Epoch-wise performance of the model on three metrics (PW-F1, AF-F1, mAP), illustrating the contributions of CGS and PGI. Early in training (epochs 0–10), the CGS-only model ("w/o PGI, with CGS", green dashed) converges faster, emphasizing event presence. After 15 epochs, the PGI-only model ("w/o CGS, with PGI", orange dashed) surpasses CGS, achieving higher final performance by refining predictions based on positional information. This demonstrates the complementary roles of CGS and PGI in accelerating early learning and improving final outcomes.

## 5 CONCLUSION AND FUTURE WORK

We presented a framework for detecting event types and boundaries in sparse time-series data by combining a global context explorer, local detail inspector, and adaptive gating module (AGM). Within the AGM, the positional Gaussian injection and conditional gate scaler provide complementary cues, improving sparse-event detection. Our approach outperforms DETR-based baselines on healthcare datasets, enabling more accurate interval identification of clinically relevant events. In real-world scenarios, the proposed framework can be applied to continuous monitoring tasks by detecting events within fixed-length segments and then merging them to reconstruct full-length events, facilitating practical and scalable deployment (Appendix E).

However, the main limitation of this work lies in its evaluation, which was confined to healthcare datasets. As a result, the generalizability across different domains remains to be validated. Future work will address this by extending the evaluation to diverse sparse-event contexts. In addition, we employed a coarse-to-fine encoder and a DETR-style decoder to effectively detect sparse events, this design may introduce challenges for real-time service deployment. To address this, we aim to explore lighter backbones and model compression strategies. We also plan to incorporate additional types of contextual information—such as patient demographics or clinical history—to further enhance the adaptability and interpretability of the gating module. Taken together, we believe that by addressing the aforementioned limitations, our framework could be widely applicable across a variety of domains, including healthcare, finance, industrial monitoring, and beyond.

## REPRODUCIBILITY STATEMENT

The model architecture is described in the main text through equations and figures. Details of the datasets, evaluation metrics, and model parameters used for implementation are provided in the Appendix. The complete source code and scripts to reproduce our framework are available at: `https://github.com/hbumjj/CDI-TS-Event-Detection`

## ACKNOWLEDGMENTS

This work was supported by the Technology Innovation Program (RS-2025-02306270, Development of a digital healthcare device for AI diagnosis and monitoring of swallowing disorders based on EGG-sEMG-ACC complex biological signals) funded By the Ministry of Trade, Industry & Resources(MOTIR, Korea).

## LLM USAGE

For the preparation of this paper, we used a large language model (LLM) solely to refine and improve the clarity of written sentences. The LLM did not contribute to the research ideas, experimental design, data analysis, or implementation of the methods described in this paper.

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

## A  DATASET

We evaluate our framework on four datasets spanning three representative tasks: (1) arrhythmia detection, (2) emotion recognition from biosignals, and (3) human activity monitoring. Each dataset is split into training, validation, and test sets with a ratio of 8:1:1, and the number of samples is summarized in Table 5. Given the severe class imbalance across datasets, we apply stratified sampling to ensure valid evaluation.

All datasets undergo a unified pre-processing pipeline comprising signal resampling and class assignment. Resampling is necessary because the backbone model used in this study accepts inputs of fixed length. Labels are adjusted during resampling to preserve class information and maintain alignment with the signals. During data preparation, any missing values in the time-series signals were addressed through linear interpolation. If a particular signal in a multivariate time-series was entirely incomplete, the corresponding data sample would be removed from the analysis. Detailed pre-processing steps and class definitions for each dataset are provided in the following subsections.

We further employ a pre-trained foundation model (Chronos-T5 (Ansari et al., 2024)) as the backbone. Chronos-T5 has not been trained on any of the datasets used in this study, thereby preserving data independence.

Table 5: Summary of datasets employed to evaluate the proposed framework across arrhythmia detection, emotion recognition, and human activity monitoring tasks. Key statistics include dataset size, number of channels, sampling frequency, and the global mean event length (GMEL).

| Dataset | MIT-BIH | SHDB-AF | WESAD | OPP |
|---|---|---|---|---|
| Dataset size | 86,208 | 466,128 | 86,733 | 28,753 |
| Variable number | 1 | 2 | 8 | 36 |
| Sampling frequency | 256 Hz | 256 Hz | 200 Hz | 30 Hz |
| GMEL | 5.19 sec | 9.90 sec | 9.41 sec | 3.93 sec |

## A.1 MIT-BIH: MIT-BIH ARRHYTHMIA DATABASE

We utilize the MIT-BIH Arrhythmia Database (Moody & Mark, 2001), a widely used benchmark for arrhythmia detection, which contains annotations for 14 types of arrhythmic rhythms. For pre-processing, ECG signals were denoised using a 1–30 Hz band-pass filter and resampled to 256 Hz for computational efficiency (Jeon et al., 2020). The signals were segmented into 10-second windows with a 1-second stride to increase data diversity. the 10-second length was chosen because it is sufficiently long to yield promising results in identifying rhythm types and classifying arrhythmias in previous studies (Liu et al., 2022b). Since ECG signals can exhibit multiple simultaneous arrhythmias, we evaluate our framework under several scenarios: (1) 3-class: AFIB and AFL detection from normal rhythm, (2) 6-class: grouping the 14 rhythms into six categories—Normal, Atrial arrhythmia, Ventricular arrhythmia, Bradycardia/Paced/Junctional rhythm (B/P/J rhythm), Conduction disorders, and Others, and (3) 15-class: using all rhythms individually. Class distributions and category definitions for the 6-class scenario are provided in Table 6.

Table 6: Class distribution and category mapping in the MIT-BIH Arrhythmia Dataset. Each class represents a specific rhythm type: normal sinus rhythm (Normal), atrial fibrillation (AFIB), atrial flutter (AFL), supraventricular tachyarrhythmia (SVTA), atrial bigeminy (AB), idioventricular rhythm (IVR), ventricular flutter (VFL), ventricular tachycardia (VT), paced rhythm (P), sinus bradycardia (SBR), nodal rhythm (NOD), ventricular bigeminy (B), pre-excitation (PREX), second-degree atrioventricular block (BII), and ventricular trigeminy (T). The "Category" row indicates the grouping used for the 6-class evaluation scenario. The "Ratio" row shows the percentage of each class in the dataset, and the "CMEL" row reports the class-specific mean event length.

| Class | Normal | AFIB | AFL | SVTA | AB | IVR | VFL | VT | P | SBR | NOD | B | PREX | BII | T |
|---|---|---|---|---|---|---|---|---|---|---|---|---|---|---|---|
| Category | 0 | 1 | 1 | 1 | 1 | 2 | 2 | 2 | 3 | 3 | 3 | 4 | 4 | 4 | 5 |
| Ratio (%) | 73.07 | 9.18 | 0.91 | 0.24 | 0.1 | 0.17 | 0.17 | 0.25 | 7.63 | 2.08 | 0.32 | 2.90 | 0.86 | 0.81 | 1.32 |
| CMEL (sec) | - | 6.70 | 2.33 | 7.75 | 5.07 | 4.54 | 0.88 | 4.60 | 2.95 | 6.19 | 2.38 | 1.25 | 5.50 | 4.43 | 5.96 |

## A.2 SHDB-AF: JAPANESE HOLTER ECG DATABASE FOR ATRIAL FIBRILLATION

We also evaluated our framework on the SHDB-AF dataset (Tsutsui et al., 2025), a two-channel Holter ECG dataset annotated with various arrhythmia rhythms. Similar to MIT-BIH, we considered two settings: (1) a 3-class task for detecting major arrhythmias (AFIB and AFL), and (2) a 5-class task corresponding to the four arrhythmia events in the dataset. For pre-processing, ECG signals were denoised using a band-pass filter (1–30 Hz) and resampled to 256 Hz. Unlike MIT-BIH, a moving window approach was not applied, as the dataset already provides sufficient coverage; signals were segmented into 10-second segments, as in the previous dataset. The class distribution and proportions are summarized in Table 7.

## A.3 WESAD: WEARABLE STRESS AND AFFECT DETECTION

The WESAD dataset (Schmidt et al., 2018) is a publicly available database for stress and emotion recognition using wearable devices. It includes six modalities: 3-axis accelerometer, ECG, EMG, EDA, temperature, and respiration. Originally recorded at 700 Hz, we resampled the signals to 200 Hz to match our backbone model. EMG was excluded due to insufficient resolution after resampling. To increase data diversity, we applied a 10-second window (Islam & Washington, 2023) with a 1-

Table 7: Class distribution in the SHDB-AF dataset. Normal denotes normal rhythm, AFIB atrial fibrillation, AFL atrial flutter, AT atrial tachycardia, and PAT & NOD represent other supraventricular tachycardias and intranodal tachycardias.

| Class | Normal | AFIB | AFL | AT | PAT & NOD |
|---|---|---|---|---|---|
| Ratio (%) | 78.57 | 19.56 | 1.54 | 0.31 | 0.03 |
| CMEL (sec) | - | 9.91 | 9.88 | 9.61 | 9.45 |

second stride. The dataset consists of 8 subject states, and their distribution is summarized in Table 8. Since states 6, 7, and 8 lack clear labels, we grouped them as Task 1, 2, and 3 for this study.

Table 8: Class Distribution in WESAD Dataset

| Class | Transient | Baseline | Stress | Amusement | Meditation | Task 1 | Task 2 | Task 3 |
|---|---|---|---|---|---|---|---|---|
| Ratio (%) | 45.44 | 20.29 | 11.48 | 6.42 | 13.60 | 0.95 | 0.91 | 0.91 |
| CMEL (sec) | - | 9.61 | 9.45 | 9.23 | 9.62 | 7.61 | 8.06 | 7.53 |

## A.4 OPP: OPPORTUNITY ACTIVITY RECOGNITION

The OPPORTUNITY (OPP) dataset (Roggen et al., 2010) is a publicly available dataset designed for human activity recognition research. It contains multimodal recordings from wearable and environmental sensors, including inertial measurement units (IMU), accelerometers, and position sensors, collected in various daily-life scenarios. For our study, we selected 36 representative accelerometer signals from the dataset for analysis. Signals were used at their original sampling frequency (30Hz) without resampling. To enhance data diversity, we applied a sliding window approach with a window length of 10 seconds and a stride of 1 second. Although human activity recognition (HAR) datasets typically use short windows (Jaén-Vargas et al., 2022), during our experiments we found that such short windows tend to perform more like classification rather than event detection. Therefore, we adopted a relatively long window of 10 seconds (Mekruksavanich & Jitpattanakul, 2022; Duan et al., 2023), as used in some HAR studies. Table 9 lists the selected accelerometer signals along with their sensor locations. The OPP dataset provides annotations at multiple hierarchical levels. In this study, we focused on the highest-level labels, corresponding to five distinct activity classes. These are summarized in Table 10.

Table 9: Selected accelerometer signals from the OPP dataset. Superscripts "$^\wedge$" and "$_\lrcorner$" denote sensors at different positions on the same limb.

| Sensor Location | Signals (X, Y, Z) |
|---|---|
| Right Knee (RKN$^\wedge$) | accX, accY, accZ |
| Hip (HIP) | accX, accY, accZ |
| Left Upper Arm (LUA$^\wedge$) | accX, accY, accZ |
| Right Upper Arm (RUA$_\lrcorner$) | accX, accY, accZ |
| Left Hand (LH) | accX, accY, accZ |
| Back (BACK) | accX, accY, accZ |
| Right Knee (RKN$_\lrcorner$) | accX, accY, accZ |
| Right Wrist (RWR) | accX, accY, accZ |
| Right Upper Arm (RUA$^\wedge$) | accX, accY, accZ |
| Left Upper Arm (LUA$_\lrcorner$) | accX, accY, accZ |
| Left Wrist (LWR) | accX, accY, accZ |
| Right Hand (RH) | accX, accY, accZ |

Table 10: Class Distribution in OPP Dataset

| Class | Base | Stand | Walk | Sit | Lie |
|---|---|---|---|---|---|
| Ratio (%) | 14.71 | 42.36 | 24.16 | 15.93 | 2.84 |
| CMEL (sec) | - | 4.03 | 2.78 | 7.29 | 6.80 |

# B  METRIC

In this study, we employed three evaluation metrics: point-wise F1 score (PW-F1), affiliation F1 score (AF-F1), and mean average precision (mAP).

**The point-wise F1 score (PW-F1)** was computed by transforming the predicted events, defined by their start point, end point, and class label, into a one-dimensional time-series sequence and comparing it with the ground-truth sequence. Let the sequence length be $T$. For each time point $t \in \{1, \ldots, T\}$, the predicted label $\hat{y}_t$ and the ground-truth label $y_t$ were compared point by point to calculate the number of true positives (*TP*), false positives (*FP*), and false negatives (*FN*) for each class. Then, the F1-score was computed for each class, and the final macro point-wise F1 score was obtained by averaging across all classes equally. This metric evaluates the overall prediction accuracy across the entire time series.

$$\text{PW-F1}_c = \frac{2 \cdot TP_c}{2 \cdot TP_c + FP_c + FN_c} \tag{14}$$

$$\text{PW-F1} = \frac{1}{C} \sum_{c=1}^{C} \text{PW-F1}_c \tag{15}$$

**The affiliation F1 score (AF-F1)** evaluates model performance at the event level. Unlike point-wise metrics, which measure accuracy at each time step, AF-F1 quantifies how well predicted event segments align with ground-truth segments, serving as a complementary metric to the point-wise F1 score. An event is defined by a start time, an end time, and a class label. A predicted event is considered a true positive (*TP*) if it belongs to the same class as a ground-truth event and its temporal span overlaps with the ground-truth segment. A false negative (*FN*) occurs when a ground-truth event is not predicted, and a false positive (*FP*) occurs when a predicted event does not overlap with any ground-truth event. The AF-F1 score is computed per class as

$$\text{AF-F1}_c = \frac{2 \cdot TP_c}{2 \cdot TP_c + FP_c + FN_c}, \tag{16}$$

and averaged across classes using a macro scheme:

$$\text{AF-F1} = \frac{1}{C} \sum_{c=1}^{C} \text{AF-F1}_c. \tag{17}$$

This metric provides a more meaningful assessment for time-series event detection by evaluating the accuracy of predicted events in terms of both their class and temporal boundaries, accounting for cases where a single event may be split into multiple events may be merged into one.

**The mean average precision (mAP)** measures model performance by first computing the area under the precision-recall (*PR*) curve for each class, referred to as average precision (*AP*), and then averaging the APs across all classes. Originally widely used in object detection and DETR frameworks, mAP is employed here as an auxiliary metric for evaluating event detection in time-series data. Here, each event is treated as a temporal segment, and the PR curve is derived from the confidence scores of predicted events. As such, mAP captures not only the correctness of event predictions but also the quality of their ranking based on prediction confidence, offering a comprehensive assessment of overall model performance.

$$\text{AP}_c = \int_0^1 P_c(R) \, dR \tag{18}$$

$$\text{mAP} = \frac{1}{C} \sum_{c=1}^{C} \text{AP}_c \tag{19}$$

## C  IMPLEMENTATION DETAILS

### C.1  MODEL ARCHITECTURE

Our framework builds upon Chronos's tiny-bolt architecture (Ansari et al., 2024) and comprises three primary components: (1) a feature extractor, (2) an AGM, and (3) a query-based decoder for event detection. The feature extractor consists of the GCE and the LDI. GCE captures long-range temporal dependencies, while LDI extracts local temporal patterns. The AGM contains two sub-modules: the CGS, which adjusts feature importance based on event presence, and the PGI, which encodes positional information of events. An overall forward-pass algorithm flow is provided in Algorithm 1, and detailed descriptions of each module, including their dimensions, are summarized in Table 11. The model has approximately 11.0M–13.3M parameters depending on the number of input channels, making it relatively compact. Considering the sparsity of events in time-series data, we set the number of queries to 10 during training. The GCE and LDI use kernel sizes of 7 and 3, respectively, and the feature dimension $d$ was experimentally set to 40 for optimal performance. Full architectural details, including layer dimensions, kernel sizes, and other hyperparameters, are provided in our GitHub repository (`https://github.com/hbumjj/CDI-TS-Event-Detection`).

---

**Algorithm 1** Overall Model Forward Flow

---

1: **procedure** FORWARDPASS(Time Series Input $X \in \mathbb{R}^{B \times T \times D}$)
2:                                                              ▷ Chronos Embedding and Initial Projection
3:       $\text{Features}_d \leftarrow \text{ChronosEmbed}(X_{:,:,d})$ for $d$ in $1..D$
4:       $h \leftarrow \text{FFN\_Block}(\text{Concatenate}(\text{Features}_d))$
5:       $h_{\text{align}} \leftarrow \text{AlignmentLayer}(h)$
6:                                                              ▷ GCE: Global Context Explorer ($f_{\text{GCE}}$)
7:       $GCE \leftarrow \text{GCE\_Block}(h_{\text{align}})$
8:       $GCE_{\text{align}} \leftarrow \text{AlignmentLayer}(GCE)$
9:                                                              ▷ LDI: Local Detail Inspector ($f_{\text{LDI}}$)
10:      $x_{\text{LDI}} \leftarrow \text{Concatenate}(h_{\text{align}}, GCE_{\text{align}})$
11:      $LDI \leftarrow \text{LDI\_Block}(x_{\text{LDI}})$
12:      $LDI_{\text{align}} \leftarrow \text{AlignmentLayer}(LDI)$
13:                                                              ▷ AGM: Adaptive Gating Module
14:      $x_{\text{AGM}} \leftarrow \text{CrossAttention}(LDI_{\text{align}}, GCE_{\text{align}})$
15:      $w_c \leftarrow \text{CGS\_Block}(GCE_{\text{align}})$
16:      $\tilde{x}_{\text{AGM}} \leftarrow w_c \odot x_{\text{AGM}}$
17:      $Output_{\text{PGI}} \leftarrow \text{Concatenate}(x_{\text{AGM}}, \text{Conv}(\tilde{x}_{\text{AGM}}))$
18:      $g \leftarrow \text{Sigmoid}(\text{Conv}(output_{\text{PGI}}))$
19:      $h_{\text{gated}} \leftarrow g \odot LDI_{\text{align}} + (1 - g) \odot GCE_{\text{align}}$
20:                                                              ▷ DETR-Style Detection Head
21:      $\text{Decoder}_{\text{out}} \leftarrow \text{TransformerDecoder}(\text{Target} = \text{Query}_{\text{embed}}, \text{Memory} = h_{\text{gated}})$
22:      $\text{Pred}_{\text{boxes}} \leftarrow \text{BBoxPredictor}(\text{Decoder}_{\text{out}})$
23:      $\text{Pred}_{\text{logits}} \leftarrow \text{ClassPredictor}(\text{Decoder}_{\text{out}})$
24:      **return** $\{\text{Pred}_{\text{boxes}}, \text{Pred}_{\text{logits}}\}$
25:
26: **end procedure**

---

### C.2  TRAINING PROCEDURE AND HYPERPARAMETERS

The model was trained for 100 epochs with a batch size of 64 and a learning rate of $1 \times 10^{-4}$. Early stopping was applied, halting training if no improvement occured for five consecutive evaluations. The dataset was split into training, validation, and test sets in an 8:1:1 ratio using stratified sampling to preserve class distributions. We used the AdamW optimizer with a weight decay of $5 \times 10^{-2}$, $\epsilon = 1 \times 10^{-8}$, and $\beta = (0.9, 0.999)$. The overall loss combined three components: $L_{\text{cos}}$, $L_{\text{BCE}}$, and $L_{\text{Detection}}$, with relative weights 2:1:7. For the Hungarian matching in detection, the class, center, and length costs were weighted 1:5:1, respectively. The Chronos model dimension was increased from 2048 to 4096. Random seeds were fixed to 42 to ensure reproducibility across all experiments.

Table 11: Overview of the proposed model architecture.

| Module | Purpose | Input / Output | Description |
|---|---|---|---|
| ChronosEmbed | Per-channel feature encoding | $[B, T, D] \rightarrow [B, \tau, D \times d_{\mathrm{FM}}]$ | Pre-trained embedding for each channel |
| FFN_Block | Compress the embedding dimension | $[B, \tau, D \times d_{\mathrm{FM}}] \rightarrow [B, \tau, d]$ | Linear projection with normalization and positional embedding |
| GCE_Block | Capture global temporal features | $[B, \tau, d] \rightarrow [B, \tau, d]$ | Transformer encoder with TCN-attention and positional embedding |
| LDI_Block | Capture local temporal features | $[B, \tau, 2d] \rightarrow [B, \tau, d]$ | Transformer encoder with TCN-attention and positional embedding |
| AGM (CGS + PGI) | Adaptive interaction | $[B, \tau, d] \rightarrow [B, \tau, 1]$ | Adjusts features based on event presence and positional encoding |
| Decoder | Query-based event prediction | $[B, N, d] \rightarrow [B, N, d]$ | Transformer decoder with learnable queries |
| Predictor | Event localization/classification | $[B, N, d] \rightarrow [B, N, 2/num\_classes]$ | Linear layers to predict event boundaries and labels |

### C.3 HARDWARE AND SOFTWARE SETUP

All experiments were conducted on a machine running Ubuntu 22.04, equipped with two Intel Xeon Scalable 6526Y processors (16 cores, 32 threads) and 256 GB DDR5 ECC RAM. Three NVIDIA Quadro RTX A5000 GPUs (24 GB GDDR6) were used for all training and evaluation. Experiments were executed within a Docker container using Python 3.10 and PyTorch 2.3.0 with CUDA 11.8.

## D COMPARISON OF CONVERGENCE SPEED WITH BASELINE MODELS

Although the primary goal of our model is improved sparse event detection in time-series data, it also demonstrates faster convergence compared to baseline models. To illustrate this, we compare the convergence behaviors up to 40 epochs (Figure 6), as most models terminate around this point due to early stopping. Specifically, we present the results for the mAP metric as a representative example.

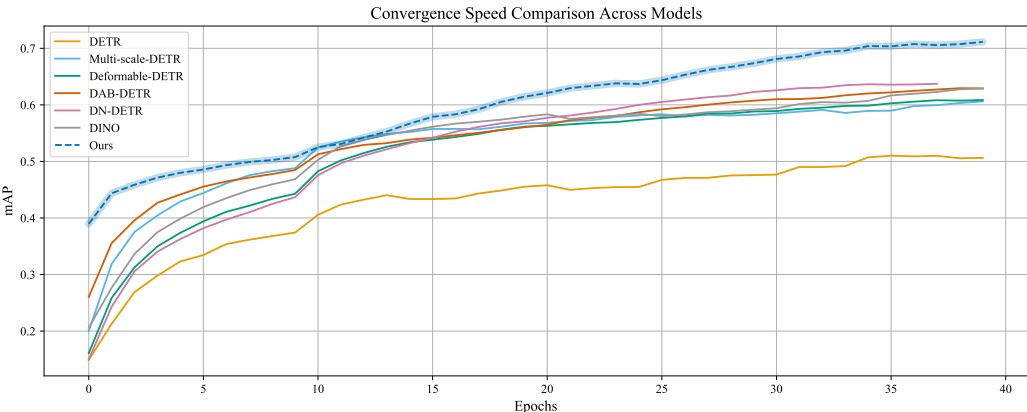

Figure 6: Convergence comparison of our model against baseline models, including DETR, Multi-scale DETR, Deformable DETR, DAB-DETR, DN-DETR, and DINO. The plot shows mAP over training epochs. Our model consistently achieves higher mAP and reaches peak performance in fewer epochs, indicating both improved effectiveness and training efficiency.

## E SAMPLE PREDICTIONS

We present representative examples of our framework's event detection results across multiple time-series datasets. For clarity, the signals are normalized within each class range, and selected cases are shown to illustrate overall trends (from Figure 7 to Figure 13). Each figure compares the model's

predictions (Blue) with the ground truth (Red). The top waveform shows the original signal, while the two colored traces below indicate the occurrence of specific events.

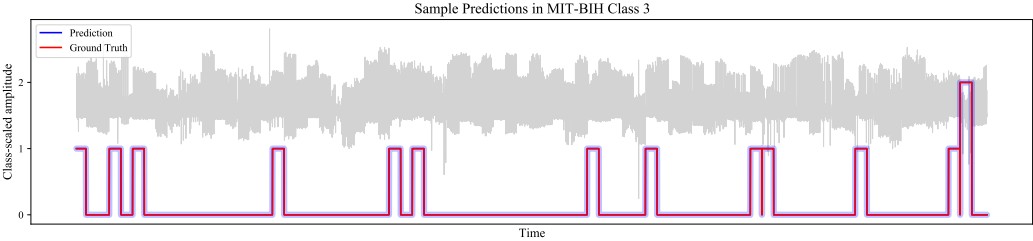

Figure 7: Sample Predictions in MIT-BIH Class 3.

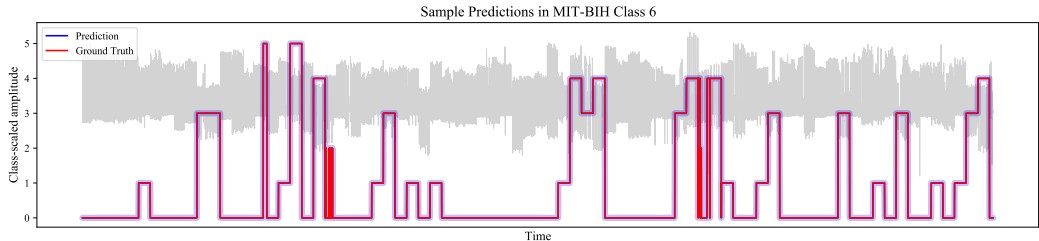

Figure 8: Sample Predictions in MIT-BIH Class 6.

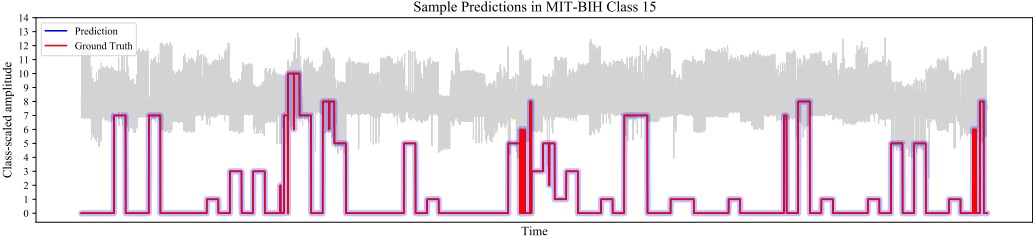

Figure 9: Sample Predictions in MIT-BIH Class 15.

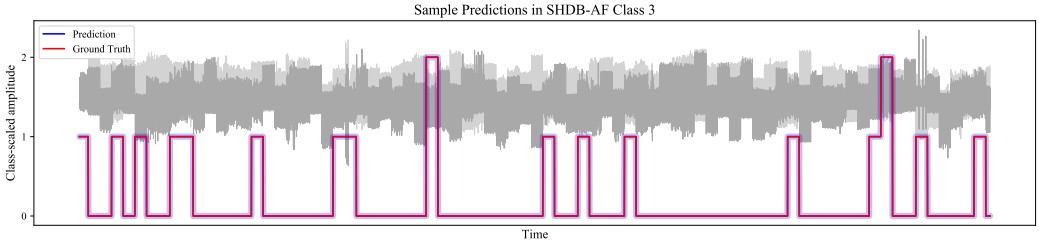

Figure 10: Sample Predictions in SHDB-AF Class 3.

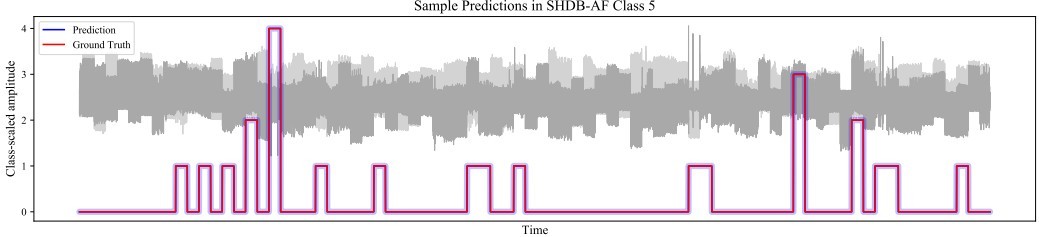

Figure 11: Sample Predictions in SHDB-AF Class 5.

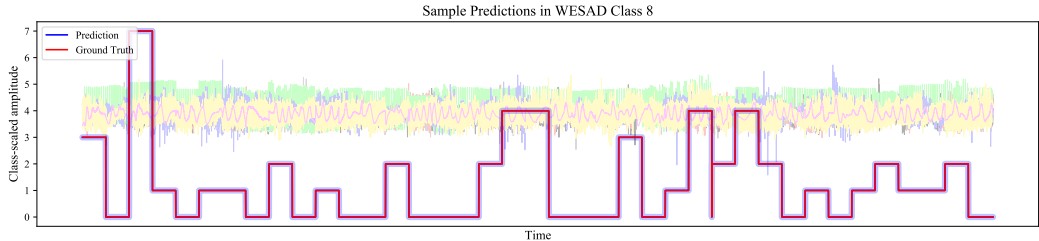

Figure 12: Sample Predictions in WESAD Class 8.

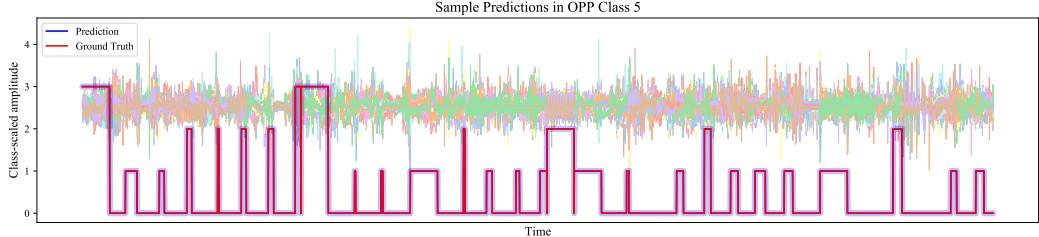

Figure 13: Sample Predictions in OPP Class 5.

# F  FULL EXPERIMENTAL RESULTS

We report the full experimental results, including per-class PW-F1 and AF-F1 scores, for all datasets from Table 12 to Table 18. A comprehensive summary and key results are presented in Section 4.1.

Table 12: Full Results: MIT-BIH 3 classes

| Class | Ratio(%) | Metric | DETR | Multi-scale DETR | Deformable DETR | DAB-DETR | DN-DETR | DINO | Ours |
|-------|----------|--------|------|------------------|-----------------|----------|---------|------|------|
| Base | 89.91 | PW-F1 | 98.86 | 99.44 | 99.34 | 99.46 | 98.84 | 99.48 | 99.72 |
|      |          | AF-F1 | - | - | - | - | - | - | - |
| AFIB | 9.18 | PW-F1 | 90.42 | 93.08 | 93.61 | 93.95 | 88.99 | 94.81 | 96.67 |
|      |          | AF-F1 | 88.04 | 91.49 | 87.51 | 90.00 | 84.44 | 87.31 | 87.61 |
| AFL | 0.91 | PW-F1 | 64.02 | 58.11 | 77.17 | 72.30 | 66.65 | 78.01 | 84.58 |
|      |          | AF-F1 | 77.39 | 60.51 | 81.90 | 77.39 | 71.04 | 79.13 | 84.30 |

Table 13: Full Results: MIT-BIH 6 classes

| Class | Ratio(%) | Metric | DETR | Multi-scale DETR | Deformable DETR | DAB-DETR | DN-DETR | DINO | Ours |
|---|---|---|---|---|---|---|---|---|---|
| Normal | 73.07 | PW-F1 | 93.81 | 95.99 | 95.45 | 94.77 | 95.08 | 96.18 | 97.46 |
| | | AF-F1 | - | - | - | - | - | - | - |
| Atrial arrhythmia | 10.43 | PW-F1 | 86.82 | 91.08 | 91.57 | 87.78 | 89.99 | 91.93 | 95.23 |
| | | AF-F1 | 74.78 | 74.95 | 76.90 | 78.29 | 75.88 | 77.32 | 78.53 |
| Ventricular arrhythmia | 0.59 | PW-F1 | 20.26 | 44.59 | 29.88 | 44.51 | 24.53 | 54.71 | 79.04 |
| | | AF-F1 | 28.27 | 36.36 | 30.83 | 25.27 | 21.75 | 25.60 | 19.52 |
| B/P/J rhythm | 10.03 | PW-F1 | 95.26 | 95.76 | 95.18 | 95.73 | 94.95 | 96.10 | 96.51 |
| | | AF-F1 | 82.05 | 79.49 | 79.52 | 82.35 | 78.46 | 79.02 | 82.63 |
| Conduction disorders | 4.57 | PW-F1 | 73.54 | 72.91 | 70.22 | 68.18 | 70.99 | 74.48 | 84.83 |
| | | AF-F1 | 59.05 | 62.29 | 59.28 | 64.62 | 57.91 | 61.04 | 68.91 |
| Others | 1.32 | PW-F1 | 44.60 | 62.45 | 56.55 | 59.03 | 53.80 | 53.31 | 61.23 |
| | | AF-F1 | 47.30 | 58.85 | 57.52 | 53.30 | 56.88 | 57.02 | 54.75 |

Table 14: Full Results: MIT-BIH 15 classes

| Class | Ratio(%) | Metric | DETR | Multi-scale DETR | Deformable DETR | DAB-DETR | DN-DETR | DINO | Ours |
|---|---|---|---|---|---|---|---|---|---|
| Normal | 73.07 | PW-F1 | 87.89 | 95.05 | 91.95 | 93.17 | 93.48 | 94.50 | 96.36 |
| | | AF-F1 | - | - | - | - | - | - | - |
| AFIB | 9.18 | PW-F1 | 78.43 | 91.38 | 83.29 | 84.54 | 86.66 | 89.34 | 91.60 |
| | | AF-F1 | 63.01 | 73.26 | 68.71 | 70.84 | 70.32 | 71.38 | 74.24 |
| AFL | 0.91 | PW-F1 | 7.25 | 49.08 | 31.94 | 19.46 | 42.82 | 33.24 | 64.75 |
| | | AF-F1 | 12.52 | 33.67 | 22.66 | 29.36 | 29.20 | 23.47 | 18.53 |
| SVTA | 0.24 | PW-F1 | 81.21 | 96.71 | 94.72 | 96.36 | 94.24 | 96.43 | 96.34 |
| | | AF-F1 | 84.53 | 85.71 | 85.42 | 88.42 | 85.22 | 87.11 | 88.45 |
| AB | 0.10 | PW-F1 | 91.82 | 81.67 | 85.02 | 74.59 | 97.08 | 95.61 | 97.01 |
| | | AF-F1 | 60.99 | 52.44 | 54.39 | 53.49 | 61.06 | 58.52 | 62.21 |
| IVR | 0.17 | PW-F1 | 56.91 | 66.43 | 61.84 | 48.49 | 65.02 | 61.62 | 68.85 |
| | | AF-F1 | 56.18 | 49.40 | 48.76 | 43.89 | 53.36 | 49.26 | 57.81 |
| VFL | 0.17 | PW-F1 | 3.36 | 31.54 | 31.53 | 22.68 | 16.55 | 32.16 | 69.03 |
| | | AF-F1 | 15.37 | 17.43 | 13.27 | 22.93 | 9.38 | 11.69 | 10.27 |
| VT | 0.25 | PW-F1 | 17.20 | 50.29 | 43.95 | 38.48 | 32.80 | 41.18 | 46.29 |
| | | AF-F1 | 28.09 | 43.06 | 43.74 | 38.01 | 35.12 | 39.36 | 46.23 |
| P | 7.63 | PW-F1 | 6.63 | 69.51 | 71.55 | 63.31 | 70.26 | 70.60 | 80.05 |
| | | AF-F1 | 37.60 | 44.64 | 50.83 | 51.91 | 50.25 | 51.15 | 55.79 |
| SBR | 2.08 | PW-F1 | 83.52 | 93.48 | 93.66 | 95.65 | 93.42 | 94.44 | 89.46 |
| | | AF-F1 | 70.71 | 74.87 | 73.30 | 75.56 | 75.68 | 75.39 | 67.96 |
| NOD | 0.32 | PW-F1 | 27.49 | 44.10 | 31.31 | 21.93 | 40.99 | 33.63 | 40.98 |
| | | AF-F1 | 24.05 | 30.97 | 31.11 | 15.95 | 33.72 | 26.52 | 34.42 |
| B | 2.90 | PW-F1 | 2.14 | 43.97 | 30.83 | 30.95 | 29.45 | 34.88 | 48.68 |
| | | AF-F1 | 2.09 | 13.13 | 14.50 | 57.14 | 11.31 | 14.59 | 16.59 |
| PREX | 0.86 | PW-F1 | 64.86 | 80.80 | 70.54 | 76.71 | 65.37 | 62.03 | 81.12 |
| | | AF-F1 | 59.57 | 66.67 | 48.78 | 49.23 | 41.67 | 50.00 | 65.00 |
| BII | 0.81 | PW-F1 | 57.07 | 80.40 | 61.13 | 47.82 | 76.80 | 82.81 | 93.72 |
| | | AF-F1 | 58.82 | 61.02 | 53.97 | 32.26 | 53.97 | 63.49 | 63.83 |
| T | 1.32 | PW-F1 | 0.60 | 38.31 | 34.19 | 28.06 | 58.97 | 60.94 | 80.17 |
| | | AF-F1 | 33.33 | 44.44 | 30.00 | 32.26 | 52.94 | 58.33 | 78.57 |

Table 15: Full Results: SHDB-AF 3 classes

| Class | Ratio(%) | Metric | DETR | Multi-scale DETR | Deformable DETR | DAB-DETR | DN-DETR | DINO | Ours |
|---|---|---|---|---|---|---|---|---|---|
| Base | 78.91 | PW-F1 | 98.83 | 99.23 | 99.03 | 98.96 | 98.77 | 99.06 | 99.48 |
| | | AF-F1 | - | - | - | - | - | - | - |
| AFIB | 19.56 | PW-F1 | 95.09 | 96.02 | 95.26 | 95.68 | 95.08 | 95.40 | 97.68 |
| | | AF-F1 | 95.75 | 96.42 | 95.81 | 96.15 | 84.95 | 94.11 | 97.84 |
| AFL | 1.54 | PW-F1 | 91.39 | 87.95 | 86.45 | 90.54 | 88.10 | 83.74 | 94.78 |
| | | AF-F1 | 91.19 | 87.85 | 86.38 | 90.31 | 87.57 | 83.54 | 94.33 |

Table 16: Full Results: SHDB-AF 5 classes

| Class | Ratio(%) | Metric | DETR | Multi-scale DETR | Deformable DETR | DAB-DETR | DN-DETR | DINO | Ours |
|---|---|---|---|---|---|---|---|---|---|
| Normal | 78.57 | PW-F1 | 97.31 | 92.53 | 97.36 | 96.59 | 97.90 | 98.51 | 99.29 |
|  |  | AF-F1 | - | - | - | - | - | - | - |
| AFIB | 19.56 | PW-F1 | 89.74 | 83.55 | 86.02 | 85.21 | 92.01 | 94.49 | 97.33 |
|  |  | AF-F1 | 90.96 | 85.08 | 87.62 | 86.74 | 91.97 | 94.80 | 97.40 |
| AFL | 1.54 | PW-F1 | 79.22 | 74.22 | 66.49 | 75.58 | 85.98 | 88.07 | 94.68 |
|  |  | AF-F1 | 79.04 | 75.66 | 66.64 | 75.49 | 84.62 | 86.96 | 94.29 |
| AT | 0.31 | PW-F1 | 42.69 | 58.20 | 38.28 | 32.81 | 65.82 | 55.68 | 75.20 |
|  |  | AF-F1 | 43.59 | 59.29 | 38.81 | 33.85 | 66.23 | 57.64 | 76.16 |
| PAT&NOD | 0.03 | PW-F1 | 29.12 | 37.80 | 22.51 | 12.50 | 43.84 | 25.85 | 66.41 |
|  |  | AF-F1 | 29.75 | 38.65 | 22.88 | 12.50 | 37.69 | 24.54 | 67.26 |

Table 17: Full Results: WESAD 8 classes

| Class | Ratio(%) | Metric | DETR | Multi-scale DETR | Deformable DETR | DAB-DETR | DN-DETR | DINO | Ours |
|---|---|---|---|---|---|---|---|---|---|
| Transient | 45.44 | PW-F1 | 71.00 | 81.41 | 80.13 | 82.53 | 75.71 | 80.44 | 85.63 |
|  |  | AF-F1 | - | - | - | - | - | - | - |
| Baseline | 20.29 | PW-F1 | 72.95 | 83.15 | 82.89 | 83.91 | 78.30 | 80.11 | 84.25 |
|  |  | AF-F1 | 75.71 | 83.92 | 84.93 | 85.52 | 77.29 | 81.59 | 85.20 |
| Stress | 11.48 | PW-F1 | 75.93 | 84.41 | 85.44 | 83.63 | 77.02 | 85.28 | 86.58 |
|  |  | AF-F1 | 77.45 | 84.94 | 85.79 | 83.70 | 73.76 | 85.33 | 86.57 |
| Amusement | 6.42 | PW-F1 | 51.15 | 66.65 | 70.81 | 76.53 | 57.49 | 65.79 | 62.50 |
|  |  | AF-F1 | 55.39 | 68.15 | 71.34 | 77.15 | 57.78 | 68.42 | 63.40 |
| Meditation | 13.60 | PW-F1 | 89.42 | 90.54 | 90.41 | 90.96 | 89.06 | 89.09 | 88.44 |
|  |  | AF-F1 | 89.79 | 90.63 | 91.17 | 91.19 | 88.90 | 89.87 | 89.04 |
| Task 1 | 0.95 | PW-F1 | 30.11 | 31.36 | 10.41 | 37.46 | 41.56 | 49.59 | 60.00 |
|  |  | AF-F1 | 34.99 | 31.69 | 12.57 | 44.40 | 41.51 | 50.97 | 57.12 |
| Task 2 | 0.91 | PW-F1 | 39.84 | 50.21 | 50.64 | 13.26 | 58.17 | 48.61 | 56.42 |
|  |  | AF-F1 | 44.15 | 56.38 | 55.66 | 16.80 | 62.28 | 51.50 | 63.92 |
| Task 3 | 0.91 | PW-F1 | 57.30 | 40.87 | 48.21 | 3.37 | 63.49 | 66.07 | 77.00 |
|  |  | AF-F1 | 58.55 | 43.83 | 56.72 | 11.31 | 62.75 | 67.45 | 74.78 |

Table 18: Full Results: OPP 5 classes

| Class | Ratio(%) | Metric | DETR | Multi-scale DETR | Deformable DETR | DAB-DETR | DN-DETR | DINO | Ours |
|---|---|---|---|---|---|---|---|---|---|
| Base | 14.71 | PW-F1 | 41.49 | 41.45 | 42.06 | 41.87 | 41.61 | 40.25 | 44.15 |
|  |  | AF-F1 | - | - | - | - | - | - | - |
| Stand | 42.36 | PW-F1 | 46.75 | 40.02 | 36.45 | 44.28 | 35.35 | 35.73 | 55.61 |
|  |  | AF-F1 | 45.91 | 40.61 | 32.97 | 44.53 | 27.37 | 29.18 | 48.83 |
| Walk | 24.16 | PW-F1 | 26.98 | 28.87 | 34.44 | 28.14 | 24.73 | 20.74 | 28.15 |
|  |  | AF-F1 | 39.21 | 37.57 | 45.34 | 38.99 | 35.78 | 30.62 | 36.98 |
| Sit | 15.93 | PW-F1 | 88.00 | 87.07 | 88.54 | 82.29 | 86.60 | 88.14 | 89.51 |
|  |  | AF-F1 | 76.88 | 76.22 | 78.34 | 77.45 | 76.01 | 78.13 | 77.58 |
| Lie | 2.84 | PW-F1 | 83.46 | 76.22 | 84.78 | 79.69 | 81.31 | 82.58 | 86.67 |
|  |  | AF-F1 | 85.58 | 79.13 | 87.03 | 82.54 | 85.03 | 85.27 | 87.86 |

