# OpenReview forum: "Enhancing Sparse Event Detection in Healthcare Time-Series via Adaptive Gate of Context–Detail Interaction"
_ICLR.cc/2026/Conference — ICLR 2026 Poster_

### Official Review · Reviewer_FGhW · 2025-10-29

**Soundness:** 3
**Presentation:** 3
**Contribution:** 3
**Rating:** 6
**Confidence:** 4

**Summary:**

The paper proposes a DETR-inspired framework for sparse event detection in healthcare time-series. It introduces a dual-branch encoder—a Global Context Explorer and a Local Detail Inspector—fused through an Adaptive Gating Module (AGM) that dynamically balances global and local information. To handle extreme event sparsity, the authors design Positional Gaussian Injection for soft temporal supervision and a Conditional Gate Scaler to mitigate class imbalance. Experiments on several physiological datasets show consistent improvements over DETR-based baselines, achieving more precise event localization and better data efficiency.

**Strengths:**

The paper’s novelty is moderate but meaningful. While the framework builds on DETR and existing ideas such as global–local fusion, adaptive gating, and soft temporal labels, these components are integrated with notable care. The design is elegant and well aligned with the characteristics of healthcare time-series—sparse, imbalanced, and context-dependent. The adaptive gate offers interpretable behavior, and the Gaussian label smoothing plus conditional weighting show clear practical insight. Overall, the work is not conceptually radical, but it represents a thoughtful and well-executed adaptation that brings tangible progress to sparse event detection in medical data.

The presentation is clear and well-structured, making the technical ideas easy to follow. Overall, this is a thoughtful and well-executed adaptation that combines solid engineering with practical relevance.

**Weaknesses:**

The main weaknesses are twofold.

First, the method’s generalization remains limited. It still requires dataset-specific fine-tuning, and the learned gating behavior may not transfer well across domains with different event sparsity or temporal patterns.

Second, the architecture is relatively heavy, combining dual-branch encoders, gating, and DETR-style decoding, which raises computational cost and makes real-time healthcare deployment challenging.

**Questions:**

NA

---

> ### Author Response · Authors · 2025-11-20
>
> **Dear Reviewer FGhW,**
>
> We sincerely appreciate your clear and insightful summary of our work. It captures the core contributions of our study, and we are grateful that these points were effectively communicated. Your review highlighted both strengths and areas for improvement, giving us valuable opportunities to refine our work.
> In particular, we would like to provide further clarification regarding the two weaknesses you pointed out.
>
> ---
> > #### **W1. The method’s generalization remains limited... not transfer well across domains with different event sparsity or temporal patterns.**
>
> We sincerely appreciate this valuable feedback. We agree that **domain generalization** is a limitation, and we have committed to addressing this in future work.\
> However, we emphasize that our proposed **adaptive gating module (AGM)** is specifically designed to handle **intra-domain variability** (different event sparsity and temporal patterns) effectively. As shown in **Highlights 1 & 2**, the AGM enables dynamic weighting of features, allowing the model to **adaptively focus on short/rare vs. long/frequent events across different tasks** (arrhythmia, emotion, activity) within the healthcare domain. This demonstrates a strong local adaptation capability, which is crucial for real-world application.
>
> However, we did not conduct evaluations on domains beyond the current scope. We have added a discussion on extending our evaluation to broader sparse-event contexts in the ***5. Conclusion and Future Work*** section.
>
> > ##### ***5. Conclusion and Future Work***
> > "... the main limitation of this work lies in its evaluation, which was confined to healthcare dataset. As a result, the generalizability across different domains remains to be validated. Future work will address this by extending the evaluation to diverse sparse-event contexts."
> ---
>
> > #### **W2. The architecture is relatively heavy, combining dual-branch encoders, gating, and DETR-style decoding... real-time healthcare deployment challenging.**
>
> We sincerely appreciate your valuable feedback. We acknowledge that our architecture may present challenges when deployed in real-time services.\
> The primary goal of our study was to **achieve high detection accuracy and precise boundary localization for sparse events**, which necessitated the dual-branch (coarse-to-fine) and attention-based (DETR-style) architecture. We deliberately prioritized these core objectives over raw speed
>
> Nevertheless, we agree that efficiency is also an important factor. We therefore cnducted an informal evaluation of inference speed on complex ECG signals (256 Hz, 2-channel, SHDB-AF). For 10-second segments, **our model achieves approximately 8 FPS**, which—although slower than the fastest baseline (DINO at 15 FPS)—still corresponds to **processing 10 seconds of data in only 0.125 seconds.** This is **sufficient for most continuous, non-critical health monitoring scenarios** where strict sub-second latency is not required.
> Importantly, **these measurements were obtained without any optimization** (e.g., TensorRT or ONNX), leaving room for significant speedup through model compression or deployment-oriented refinements.
>
> We fully recognize the importance of lightweight and efficient models. We have added a discussion on optimizing efficiency in the ***5. Conclusion and Future Work*** section. Thank you again for bringing attention to this critical point.
>
> > ##### ***5. Conclusion and Future Work***
> > "... In addition, we employed a coarse-to-fine encoder and a DETR-style decoder to effectively detect sparse events, this design may introduce challenges for real-time service deployment. To address this, we aim to explore lighter backbones and model compression strategies."
> ---

---

### Official Review · Reviewer_HDHf · 2025-10-30

**Soundness:** 3
**Presentation:** 3
**Contribution:** 3
**Rating:** 6
**Confidence:** 3

**Summary:**

This paper introduces a lightweight deep learning model for time-series event detection with a novel evaluation metric called Affiliation F1 score (AF-F1). The proposed model builds upon Chronos's tiny-bolt architecture and consists of three primary components: a feature extractor (comprising GCE for capturing long-range temporal dependencies and LDI for extracting local temporal patterns), an Adaptive Interaction Module (AGM) containing CGS (adjusts feature importance based on event presence) and PGI (encodes positional information), and a query-based decoder for event prediction. The AF-F1 metric evaluates model performance at the event level by measuring how well predicted event segments align with ground-truth segments, considering both class labels and temporal boundaries. The model achieves a compact size of 11.0M-13.3M parameters and demonstrates strong performance on event detection tasks.

**Strengths:**

1. The introduction of AF-F1 as an event-level metric is a significant contribution, addressing a critical limitation of point-wise metrics by considering both temporal alignment and class accuracy, which is particularly valuable for event detection tasks.
2. The separation of components for different temporal scales (GCE for long-range dependencies, LDI for local patterns) demonstrates thoughtful design that effectively captures diverse temporal features.
3. The use of both AF-F1 and mAP as complementary metrics provides a more complete assessment of model performance, capturing not only event correctness but also the quality of prediction ranking based on confidence scores.

**Weaknesses:**

1. The evaluation appears to be limited to a single dataset (implied but not explicitly stated), which may not adequately demonstrate generalizability across different domains or event types.
2. There's no detailed analysis of model performance across different event types (e.g., short vs. long events, frequent vs. rare events), which could reveal important performance characteristics and limitations.

**Questions:**

No more questions

---

> ### Author Response · Authors · 2025-11-20
>
> **Dear Reviewer HDHf,**
>
> We sincerely appreciate the reviewer for carefully reading our work and providing constructive feedback. Your comments were highly valuable in helping us clearly identify both the strengths and limitations of our study, and they have greatly contributed to improving the overall quality of our research. We acknowledge the areas that were insufficiently addressed in the original manuscript and would like to provide our response and corresponding revisions below.
>
> ---
> > #### **W1. The evaluation appears to be limited to a single dataset (implied but not explicitly stated), which may not adequately demonstrate generalizability across different domains or event types.**
>
> We sincerely appreciate this valuable feedback. We must first clarify that our evaluation was conducted across **four datasets** spanning **three distinct domains**: arrhythmia detection (MIT-BIH, SHDB-AF), emotion recognition (WESAD), and activity monitoring (OPP).
> However, as the reviewer rightly pointed out, **the generalizability across vastly different types of domains has not been fully demonstrated** through experimental validation. This limitation arises from the fact that **the proposed framework was primarily optimized for the detection of sparse events within a specific domain**. As part of our future work, we plan to extend our experiments to a broader range of domains to verify the framework’s general applicability. We are grateful for this valuable suggestion, and we have revised the ***5. Conclusion and Future Work*** section of the manuscript to explicitly emphasize this limitation.
>
> > ##### ***5. Conclusion and Future Work***
> > "... However, the main limitation of this work lies in its evaluation, which was confined to healthcare datasets. As a result, the generalizability across different domains remains to be validated. Future work will address this by extending the evaluation to diverse sparse-event contexts."
>
> ---

---

> > ### Author Response · Authors · 2025-11-20
> >
> > > #### **W2. There's no detailed analysis of model performance across different event types (e.g., short vs. long events, frequent vs. rare events), which could reveal important performance characteristics and limitations.**
> >
> > We sincerely appreciate this valuable feedback. We agree that the analysis of how the model's performance varies across different event characteristics was insufficient. **While the original manuscript included performance results comparing frequently occurring and rare (sparse) events, a detailed analysis based on specific event characteristics was not adequately provided.**
> >
> > To address this issue, we incorporated a **new relative performance analysis based on event length** into the manuscript. Rather than relying on a simple short/long categorization, we performed a more rigorous evaluation by comparing the **class-specific mean event length (CMEL)** with the **global mean event length (GMEL)**. This metric was introduced to reflect the class-dependent characteristics of event duration (e.g., differences in the duration distributions of walking vs. jumping).\
> > Furthermore, evaluating performance solely with GMEL can lead to ambiguity in defining what constitutes “short” or “long” events. This can make per-event evaluation reflect both sparsity-related bias and differences due to event length. For example, events that are frequent but short and events that are rare but short would both be treated as ‘short,’ potentially causing confusion.
> >
> > To resolve this, we used both CMEL and GMEL in our analysis. Based on GMEL, we categorized events as short or long across multiple datasets and reported the corresponding performance in Table 3, with the results summarized in **Highlight 2 of Section 4** (Experiment). Full details are **Appendices A and F**.
> >
> > > ##### ***4.1 Performance Comparison on Healthcare Datasets: Highlights 2***
> > > "... The results show that the model generally achieves high performance regardless of event duration. This trend is particularly evident in the MIT-BIH Class 15 dataset, where the model outperforms baselines for both the SVTA class—whose event duration exceeds the GMEL—and the P class, which is shorter than the GMEL ..."
> > >
> > > ***Table 3: Summary of model performance across event-length***
> >
> > | Dataset (Event)             | CMEL (GMEL) | Metric  | DETR   | Multi-scale DETR | Deformable DETR | DAB-DETR | DN-DETR | DINO   | Ours   |
> > |:---------------------------:|:-----------:|:-------:|:------:|:----------------:|:---------------:|:--------:|:-------:|:------:|:------:|
> > | ...                         | ...         | ...     | ...    | ...              | ...             | ...      | ...     | ...    | ...    |
> > | **MIT-BIH Class 15 (SVTA)**     | 7.75 (5.19) | PW-F1   | 81.21  | **96.71**        | 94.72           | 96.36    | 94.24   | 96.43  | 96.34  |
> > |                             |             | AF-F1   | 84.53  | 85.71            | 85.42           | 88.42    | 85.22   | 87.11  | **88.45** |
> > | **MIT-BIH Class 15 (P)**        | 2.95 (5.19) | PW-F1   | 6.63   | 69.51            | _71.55_         | 63.31    | 70.26   | 70.60  | **80.05** |
> > |                             |             | AF-F1   | 37.60  | 44.64            | 50.83           | _51.91_  | 50.25   | 51.15  | **55.79** |
> > | ...                         | ...         | ...     | ...    | ...              | ...             | ...      | ...     | ...    | ...    |
> >
> > ---
> > We are truly grateful for the reviewer’s constructive feedback. Your comments have greatly helped us improve the clarity, completeness, and rigor of our manuscript, and we have carefully reviewed all mathematical formulations and detailed descriptions to ensure accuracy and consistency.

---

### Official Review · Reviewer_hGZQ · 2025-11-01

**Soundness:** 3
**Presentation:** 2
**Contribution:** 2
**Rating:** 4
**Confidence:** 5

**Summary:**

This paper proposes a coarse-to-fine detection framework for enhancing sparse event detection in healthcare time-series data. The framework combines a Global Context Explorer (GCE) and a Local Detail Inspector (LDI) with an Adaptive Gating Module (AGM). The AGM uses Positional Gaussian Injection (PGI) for refined temporal localization and a Conditional Gate Scaler (CGS) for adaptive rebalancing of sparse event features. The model leverages multiple label perspectives during training and employs a DETR-based architecture for joint event type and boundary prediction. Evaluated on diverse healthcare datasets (arrhythmia, emotion, activity monitoring), the proposed method demonstrates substantial performance gains, particularly for sparse events, over existing DETR-based baselines.

**Strengths:**

1.  **Targeting a Critical Problem (Sparse Event Detection in Healthcare Time-Series):** The paper addresses a highly relevant and challenging problem in healthcare—detecting rare, clinically meaningful events with precise boundaries in complex time-series data. This problem is clearly articulated as a limitation of existing methods.
2.  **DETR-based End-to-End Framework:** Adopting and adapting the DETR architecture for time-series event detection, combined with a coarse-to-fine feature extraction, offers an elegant end-to-end solution. The tailored Hungarian matching cost for temporal events is a thoughtful adaptation.
3.  **Comprehensive Evaluation:** The framework is rigorously evaluated on four diverse healthcare datasets across three different tasks (arrhythmia, emotion, activity recognition) and various class evaluation scenarios, demonstrating consistent performance gains over multiple strong DETR baselines. The use of PW-F1, AF-F1, and mAP provides a holistic view of performance.

**Weaknesses:**

1.  **Ambiguity in Time-Series Windowing and Event Truncation:** The use of fixed 10-second windows with a 1-second stride for long-duration healthcare time-series (e.g., Holter ECG) raises concerns. The paper lacks a clear explanation of how events that span across window boundaries are handled, potentially leading to event truncation or inaccurate boundary detection, which contradicts the goal of "precise event boundary detection." This is a critical aspect for reproducibility and clinical applicability.


2.  **Underspecified "Adaptive Gating Module" (AGM) Mechanism:** The core component, the AGM, specifically the precise mechanism for generating the `g` gate tensor (e.g., what layers/activation functions are applied to which input features to produce `g` as a R $\in$ B×τ×1  tensor), is not explicitly detailed. This lack of clarity hinders the understanding of a key novelty and impacts reproducibility.


3.  **Lack of Clarity in GCE/LDI "Coarse-to-Fine" Distinction and TCN Details:** The distinction between "global" (GCE) and "local" (LDI) features is primarily attributed to TCN kernel sizes (7 vs. 3). However, **without specifying the dilation rates** used in their TCN-Attention mechanisms, it's unclear if this difference is substantial enough to justify the "coarse-to-fine" claim, especially given the inherently short window length. Furthermore, the detailed operation and structure of the "FFN-based alignment layer" are insufficiently explained.

4.  **Information Dispersal and Lack of Self-Containedness:** Several crucial details, such as full metric definitions, complete quantitative results tables (beyond highlights), detailed dataset class distributions for specific class evaluation scenarios (e.g., "Class 3", "Class 6", "Class 15" for MIT-BIH), and specific hyperparameters/adaptation methods for baselines, are mostly relegated to the Appendix. This significantly impedes the reader's ability to grasp the main content and evaluate the results. Key figures like Table 1, Table 2, and Figure 2 (which is overly complex and lacks sufficient labeling for intermediate representations like `z` and `h`) are not self-contained, forcing constant reference to the appendix and hindering streamlined understanding.

5.  **Inconsistent and Undefined Notation:** Several fundamental variables used in equations, such as the batch size `B` (Eq. 2), the total number of events `N` (Eq. 1), and the subscript `b` in $w_b(t)$ (Eq. 6), are used without explicit definition early in the methodology. Furthermore, the input definition for `X` (Eq. 1) does not include the batch dimension `B`, leading to an inconsistency with later equations that use `B`.

6.  **Limited Discussion on Clinical Interpretability/Actionability:** While the abstract and introduction mention "actionable insights" and "reliable interpretation" in real-world clinical applications, the paper's analysis is predominantly quantitative. A deeper discussion or qualitative analysis on *how* the improved event detection truly translates into concrete clinical benefits (e.g., enabling earlier alerts, more precise diagnosis, or better treatment planning) would strengthen its real-world impact claim.

**Questions:**

1.  **Event Handling at Window Boundaries:** Given the 10-second windowing approach for continuous time-series, how are ground-truth events that are longer than 10 seconds or that span across window boundaries specifically annotated and handled during training and evaluation? Does this approach risk truncating events or causing ambiguities in boundary detection, especially for a framework aiming for "precise event boundary detection"?

2.  **Precise Generation of Gate Tensor `g`:** The paper states that "the AGM produces a gate tensor `g`... that controls the relative contributions of LDI and GCE" . However, the exact mechanism or specific layers (e.g., what output features, linear layers, and activation functions) through which this `g` tensor is computed from the features (e.g., after PGI and CGS) is not explicitly described. Please clarify the precise generation process of `g`.

3.  **TCN Dilations for GCE and LDI & Alignment Layer Details:** To substantiate the "coarse-to-fine" distinction between GCE and LDI, please specify the dilation rates used in their respective TCN-Attention mechanisms. This is crucial for understanding how kernel sizes 7 and 3 translate into genuinely "global" and "local" effective receptive fields.

4.  **Clarification of Notation and Consistency:** Please provide clear and explicit definitions for all variables used in equations (e.g., `N` in Eq. 1, `B` in Eq. 2) early in the methodology section. Ensure consistency in input definitions, specifically by including the batch dimension `B` in the definition of input `X` (Eq. 1). Also, clarify the meaning of the subscript `b` in $w_b(t)$ (Eq. 6).

---

> ### Author Response · Authors · 2025-11-20
>
> **Dear Reviewer hGZQ,**
>
> We sincerely thank the reviewer for their careful reading and valuable feedback. Your comments have been extremely helpful and provided us with a great opportunity to improve the clarity, readability, and overall quality of our manuscript. We provide detailed responses below and describe how the manuscript has been revised accordingly. For clarity, we have separated our responses into multiple entries corresponding to different comments.
>
> ---
> > #### **Q1. Event Handling at Window Boundaries**
>
> We greatly appreciate the reviewer’s insightful comment. As noted, our 10-second windowing approach can lead to events longer than 10 seconds being split into multiple segments. We were aware of this issue during the study, as it may make it challenging for the model to learn event start and end points.
>
> (1) **To mitigate this, we incorporated a Positional Gaussian Injection (PGI) within the Adaptive Gating Module**. This mechanism converts the labels in to a Gaussian-shaped auxiliary signal, which serves as a guidance during training. Even for events that are split across segments, the Gaussian edges align with their start and end points, providing the model with clear cues about event boundaries within fixed-length segments during training. However, we acknowledge that this aspect was not described in the original manuscript. We have now added a clear explanation in the main text: ***3.3.2 Positional Gaussian Injection***.
>
> (2) Furthermore, we recognize that the term **precise event boundary detection** could be interpreted as an overstatement. Our method detects **precise event boundary within fixed-length segments**, regardless of whether the full event spans multiple segments or is fully connected. Accordingly, we have revised the termiology in the manuscript to avoid potential misinterpretation.
>
> (3) In summary, both training and evaluation are performed on segmented events, and we employ PGI to mitigate the ambiguity in boundary detection that arises from split segments. In practical, **real-world applications, the model could first detect events at the segment level and then aggregate the segments to reconstruct the full events.** (Appendix E)
>
> > ##### ***1. Introduction***
> > (2) " ... However, such window-based formulations cannot capture exact onset and offset times, limiting their ability to perform boundary-aware event detection within fixed segments. "
> > ##### ***2. Related Work***
> > (2) " ... While these models are effective at capturing global patterns and performing point-wise prediction or regression, they face structural limitations for event boundary detection within segments. "
> > ##### ***3.3.2 Positional Gaussian Injection***
> > (1) " ... This Gaussian supervision is deliberately designed to: (i)... (iii) prevent boundary ambiguity that arises when events are split across fixed-length segments. ... "
> > ##### ***5. Conclusion and Future work***
> > (3) " ... In real-world scenarios, the proposed framework can be applied to continuous monitoring tasks by detecting events within fixed-length segments and then merging them to reconstruct full-length events, facilitating practical and scalable deployment."
>
> ---
>
> > #### **Q2. Precise Generation of Gate Tensor $g$**
>
> We sincerely appreciate the reviewer’s thoughtful question. We acknowledge that the original manuscript did not provide sufficient detail regarding the vector generation process within the adaptive gating module (AGM). In the revised version, we have added a more comprehensive explanation to clarify this component.
>
> Specifically, the global feature (GCE: $K$, $V$) and local feature (LDI: $Q$) are first integrated through cross-attention and then fed into the AGM as input tensors of shape $(B, \tau, d)$. Within the AGM, the data sequentially pass through two submodules—the conditional gate scaler (CGS) and the positional Gaussian injection (PGI)—to produce the output $output_{PGI}$ as tensors of shape $(B,\tau,d + d_{FM})$.\
> This output is then compressed to a shape of $(B, \tau, 1)$ via a convolution layer followed by a sigmoid activation, yielding the gate vector $g$. The gate vector performs element-wise multiplication with the LDI and GCE features, dynamically adjusting their relative contributions at each time step.
>
> We have incorporated these details into the main text to improve clarity. Additionally, we have comprehensively revised ***Section 3.3 and its subsections 3.3.1 and 3.3.2***, updating the shape information for both the CGS and PGI processes, correcting erroneous equations, and reorganizing the content to improve the overall logical flow. To improve the logical flow, the order of the two subsections has been swapped: ***CGS is now described in Section 3.3.1 and PGI in Section 3.3.2***.
>
> ---

---

> ### Author Response · Authors · 2025-11-20
>
> > #### **Q3. Lack of Clarity in GCE/LDI "Coarse-to-Fine" Distinction and TCN Details**
>
> First, we sincerely thank the reviewer for this insightful question. We acknowledge that the original manuscript lacked sufficient explanation in this regard, and we have added clarifications to the revised manuscript, which we summarize below.
>
> For both the GCE and LDI $TCN-Attention$ mechanisms, we used dilation rates ($d$) of ${[1, 2, 4, 8]}$. The distinction between coarse and fine arises not only from differences in receptive fields but is also determined by **two key factors**.
>
> 1) ##### ***Information Density***
> The input to LDI is defined as $x_{\mathrm{LDI}} = h_{align} \oplus f_{\mathrm{GCE}}(h_{align})$. where $h_{align}$ denotes the initial feature after alignment layer and $f_{GCE}(h_{align})$ represents the globally refined context features from GCE. By concatenating along the channel dimension, LDI operates on a higher-dimensional, denser feature map—effectively providing twice the amount of information compared to GCE. Even with the same dilation rates, LDI focuses on local interactions across this richer feature space, enabling finer-grained refinement of details.
>
> 2) ##### ***Module placement***
> The two modules are arranged sequentially, each performing clearly separated functions. GCE employs a larger kernel size (7) to capture coarse temporal dependencies, whereas LDI applies a narrower kernel to the high-dimensional input that includes the global context from GCE, allowing it to refine local details based on the overall context. Together, this design ensures the successful implementation of a coarse-to-fine strategy, providing structural justification beyond mere differences in TCN kernel size.
>
> We apologize for the underspecified description of the FFN-based alignment layer. We have clarified that this is not a full FFN but a single linear projection layer, which we have renamed the **linear alignment layer** ($d$ $\rightarrow$ $d$) to avoid confusion.\
> Its primary role is to ensure that heterogeneous features—including the initial features $h$, global features (GCE outputs), and local features (LDI outputs)—are compatible for subsequent processing by the adaptive gating module (AGM). This is achieved through two main functions:
>
> - **Dimensionality Alignment**: It aligns features produced through different operations into a consistent representation space, which is essential for the AGM's input where GCE and LDI features must be combined.
> - **Distribution Normalization**: It reduces scale and distribution mismatches introduced by the different extraction modules (GCE and LDI). Ensuring this statistical compatibility is crucial for stable training and better generalization.
>
> These clarifications and updates have been incorporated into ***Section 3.2: Global Context Explorer & Local Detail Inspector, and a relevant citations has also been added***.
>
> > ##### ***3.2 Global Context Explorer & Local Detail Inspector***
> > " a TCN-based attention mechanism with dilation rates $[1,2,4,8]$ ..."\
> > " the coarse-to-fine hierarchy ... First, LDI receives ... Second, the sequential ordering ..."\
> > " Because all features remain in the aligned space, ..."
>
> ---
>
>
> > #### **Q4. Clarification of Notation and Consistency**
>
> We sincerely appreciate your valuable feedback. As suggested, we have clarified and revised the relevant parts.
>
> First, regarding **Equation 1**, we have made the equation clearer and added definitions for $N$ and $B$. For **Equation 2**, while it was referenced in Equation 1, we have provided an additional clarification regarding the batch size $B$ to improve readability. We have also corrected typos in the equations.
>
> For **Equation 9** (Equation 6 $\rightarrow$ Equation 9), we verified that the original equation used during the early stages of our study has been updated accordingly ($y_{gaussian}$).
> Moreover, beyond the points you raised, we are carefully reviewing all equations throughout the manuscript and will continue to make corrections as needed.
>
> Thank you again for your thoughtful and constructive comments.
>
> ---

---

> > ### Author Response · Authors · 2025-11-20
> >
> > Furthermore, we respectfully provide our revisions addressing the points that were identified as weaknesses, in addition to the questions you raised.
> >
> > ---
> > > #### **W4. Information Dispersal and Lack of Self-Containedness**
> >
> > First, we sincerely apologize for the inconvenience caused by the necessary dispersal of details due to page limits.  **We have integrated crucial information into the main text as outlined below.** To further improve reader accessibility, we have also ensured that all **PDF cross-references from the main text to the Appendix are active hyperlinks**, allowing readers to jump directly to the corresponding Appendix section.
> >
> > In the main text, we have added **supplementary details in four specific areas** where the content was previously insufficient.
> >
> > > ##### ***4.1 Performance Comparison on Healthcare Datasets***
> > > **Dataset**: added details on arrhythmia detection classes, pre-processing methods, and dataset statistics.
> > > **Baseline**: clarified adaptation methods and provided additional parameter information.
> > > **Metrics**: provided a more detailed explanation of the evaluation metrics.
> > >
> > > **Figure 2**: added intermediate representations and the pathways of the simplified model.
> >
> > We have identified some missing details in the Appendix and are currently adding the necessary content. We will keep you updated on this through the global comment.
> >
> > ---
> >
> > > #### **W6. Limited Discussion on Clinical Interpretability/Actionability**
> >
> > We sincerely appreciate the reviewer’s valuable feedback. As you suggested, while our manuscript primarily focuses on quantitative analysis, we recognized the need to more strongly connect our findings to actual clinical benefits.
> >
> > Existing segment-based classification methods provide only the presence or absence of specific events, which forces clinicians to review the entire time-series data to make informed judgments, resulting in inefficiencies. In contrast, the event detection approach presented in our study provides **the onset and offset times, as well as the types of events.** This enables clinicians to focus on specific suspicious intervals, thereby **streamlining the diagnostic workflow and supporting evidence-based decision-making with higher reliability.**\
> > Based on these improvements, **actionable insights** can be derived from well-founded, clear decision evidence. We have summarized this discussion in the ***1. Introduction*** to highlight the clinical relevance of our approach.
> >
> > > ##### ***1. Introduction***
> > > " ... Moreover, providing explicit temporal and class information allows clinicians ... supporting evidence-based decision-making, and generating actionable insights"\
> > > "... Consequently, previous methods are limited in their ability ... the need for accurate, boundary-aware detection that facilitates both reliable interpreta-
> > tion and clinical actionability."
> > ---

---

### Official Review · Reviewer_xGKy · 2025-11-01

**Soundness:** 4
**Presentation:** 4
**Contribution:** 4
**Rating:** 6
**Confidence:** 2

**Summary:**

This paper presents a novel approach for enhancing sparse event detection in healthcare time-series data through the use of an adaptive gate mechanism. The proposed method combines global context exploration, local detail inspection, and an adaptive gate module to improve the precision of event localization in medical datasets, such as ECG recordings. The authors have conducted experiments on various healthcare datasets and demonstrate substantial improvements over existing methods in terms of sparse event detection performance.

**Strengths:**

The introduction of the adaptive gate mechanism (AGM) is a significant contribution to sparse event detection. By leveraging both global and local perspectives, the method addresses key challenges in event localization. The explanation of the proposed framework and the AGM module is clear and well-structured. The authors provide sufficient details on the model's design and operational principles. The experiments on diverse healthcare datasets, including ECG signals, are comprehensive and effectively demonstrate the superiority of the proposed method over existing approaches. The results are promising and support the validity of the approach.

**Weaknesses:**

How does the proposed method handle cases with missing or incomplete data, particularly in sparse time-series recordings?
Could the adaptive gate mechanism be further enhanced by incorporating additional types of context information (e.g., patient demographics, clinical history)?

**Questions:**

How does the proposed method handle cases with missing or incomplete data, particularly in sparse time-series recordings?
Could the adaptive gate mechanism be further enhanced by incorporating additional types of context information (e.g., patient demographics, clinical history)?

---

> ### Author Response · Authors · 2025-11-20
>
> **Dear Reviewer XGKy,**
>
> We sincerely thank the reviewer for the careful and thoughtful evaluation of our work. Your insightful comments have provided us with an excellent opportunity to revisit our study and further improve the clarity and completeness of our manuscript. We provide our responses to your comments below.
>
> ---
> > #### **Q1. "How does the proposed method handle cases with missing or incomplete data, particularly in sparse time-series recordings?"**
>
> We appreciate the reviewer pointing out this important aspect. We acknowledge that our original manuscript did not explicitly describe how **missing or incomplete data** were handled.\
> In our study, missing values—which occured only sparsely and at a very low frequency—were addressed through **linear interpolation before feeding the fixed-length window** to the model. Additionally, if a particular signal in a multivariate time-series was entirely incomplete, **the corresponding data sample was removed from the analysis**.
>
> To clarify this, we have added detailed descriptions of data handling in both ***4. Experiment: Datasets*** and ***Appendix A: Dataset***, and addressed other previously missing details in the manuscript.
>
>
> > ##### ***4. Experiment: Datsets***
> > "... Sampling frequency and channel count vary: MIT-BIH and SHDB-AF are 256Hz with 1 and 2 channels; WESAD is 200Hz with 8 channels; OPP is 30Hz with 36 channels. A fixed 10-second window is applied across datasets.
> Time-series signals undergo resampling, interpolation of missing values."
> > ##### ***Appendix A: Dataset***
> > "... During data preparation, any missing values in the time-series signals were addressed through linear interpolation. If a particular signal in a multivariate time-series was entirely incomplete, the corresponding data sample would be removed from the analysis."
> ---
>
> > #### **Q2 "Could the adaptive gate mechanism be further enhanced by incorporating additional types of context information (e.g., patient demographics, clinical history)?"**
>
> We sincerely appreciate this insightful suggestion.\
> We also considered this insightful suggestion during the initial stages of our research. Unfortunately, the datasets we used **lacked complete additional context information**, so we were unable to explore this direction empirically. Nevertheless, we believe that incorporating **such information could further improve the performance of our method**.
>
> Methodologically, one could design a separate encoder to process additional context information and integrate it with the adaptive gate mechanism via a fusion layer. This would allow the model to dynamically assess the importance of each sample while taking contextual information into account. We agree that this represents a promising avenue for future work and have reflected this discussion in the ***5. Conclusion and Future Work*** section of the revised manuscript.
>
> > ##### ***5. Conclusion and Future Work***
> > "... We also plan to incorporate additional types of contextual information—such as patient demographics or clinical history—to further enhance the adaptability and interpretability of the gating mechanism."
> ---
> We are truly grateful for the reviewer’s constructive feedback. Your comments have greatly helped us improve the clarity, completeness, and rigor of our manuscript, and we have carefully reviewed all mathematical formulations and detailed descriptions to ensure accuracy and consistency.

---

### Author Response · Authors · 2025-11-24
**Minor edits  and  improved reproducibility**

**Dear reviewers,**

Thank you for your valuable feedback. We are continuously working to improve reproducibility and readability, and we would like to provide an overview of the **minor updates** and refinements made based on recent findings.

---
> ##### ***4.1 Performance Comparison on Healthcare Datasets***
> " Table 1: Overall performance comparison across multiple datasets … shown in Appendix E "

We identified that the referenced appendix was incorrect and have updated it from Appendix D to Appendix E.

> ##### ***5. Conclusion and Future Work***
> " In real-world scenarios, the proposed framework … (Appendix E) "

We have added the link to Appendix E to facilitate a clearer understanding of the description.

> ##### ***Appendix C.1 Model Architecture***
> " An overall forward-pass algorithm flow is provided in Algorithm 1 ... "

To maximize reproducibility, we have provided the model’s forward pass in a pseudo-code format. Additionally, for improved readability, we have adjusted the placement of the corresponding paragraph within Appendix C.

---

---

> ### Author Response · Authors · 2025-12-01
>
> > ##### ***Figure 2. Overview of the proposed framework.***
> > "The GCE and LDI outputs are fused through the adaptive gating module (AGM), which acts as a dynamic gate to modulate global and local information."
>
> To emphasize the meaning, “integrate” was replaced with “modulate.”
>
>
> > ##### ***Figure 5. Epoch-wise performance of the model on three metrics ...***
> > **Typo corrected:** ‘w/o PGI, with CGS ’ → “w/o PGI, with CGS”
>
> ---

---

### Author Response · Authors · 2025-12-02

***Dear Area Chair,***

We have made extensive efforts to incorporate the **valuable comments from the reviewers** and improve the readability of the manuscript. At the same time, **we ensured that the core content of the paper remains intact throughout the revision process,** so that the research itself is not fundamentally altered or perceived as an entirely new study.

The main types of weaknesses pointed out by the reviewers can be categorized into two areas:

---

>#### ***1.  Ambiguity in explanations and insufficient information***
> The weaknesses and questions raised by the reviewers mainly stemmed from unclear explanations or a lack of sufficient information. Accordingly, **we have extensively revised Section 3: Methodology, reorganized figures and equations for clarity, and presented additional results to enhance comprehensibility.** Furthermore, we have included a pseudo-code representation of our proposed framework in the appendix. We believe that these revisions will effectively address the reviewers’ concerns and questions.

>#### ***2.  Limited validation beyond healthcare domains***
> **The primary objective of our study is to evaluate how effectively events can be detected, particularly focusing on sparse class events (e.g., diseases in healthcare).**  Consistent with conventional DETR-family models, our methodology likewise focused on maximizing detection performance for specific, pre-defined classes in the target dataset. Broader goals like zero-shot or inference speed were intentionally set aside as they fall outside the scope of this study. Although these points were raised as limitations during the revision process, they were not the main focus of this research. Nevertheless, we recognize them as important insights and potential directions for future work. Accordingly, **we revised the manuscript to more clearly emphasize the core objectives and to outline possible future directions in more detail.**

---

In accordance with the reviewers’ requests, all revised sections have been highlighted in ***violet***. Finally, in response to the evolving review environment, we continuously reviewed the manuscript to correct minor errors throughout the process.

Thank you for your kind consideration.

---

### Meta-Review · Area_Chair_UUh6 · 2026-01-07

**Summary:**

Most of raised concerns focus on design rationales and experiment setups, which have been generally addressed with further clarifications. Limitations in terms of evaluation on other domains as well as alternative designs are acknowledged but they did not affect reviewers’ overall positive ratings.

**Reviewer Concerns:**

`xGKy`: Handling missing/incomplete data (addressed with clarifications); alternative designs on adaptive gating with additional contexts (left for future work)

`hGZQ`: Cross-window event handling, lack of details / clarity issues on key designs, inconsistent notations, limited discussions on clinical interpretability (generally addressable with additional details and revisions)

`HDHf`: Narrow evaluations, missing breakdown of performance over event types (misinterpretation clarified, performance breakdown added to the Appendix)

`FGhW`: Generalization to other domains/patterns (partly clarified in terms of addressing intra-domain variability and promised future works); heavy network design (additional information on FPS and further avenues to acceleration mentioned)

**Reviewer Scores:**

Most raised concerns are over clarity of designs and experiments. Further clarifications have been made by the authors. It is therefore reasonable to postulate that most reviewers would remain overall positive.

---

### Decision · Program_Chairs · 2026-01-26

Accept (Poster)